# Intestinal fungi are causally implicated in microbiome assembly and immune development in mice

Erik van Tilburg Bernardes [1,2,3,4,5], Veronika Kuchařová Pettersen [1,2,3,4,5,6], Mackenzie W. Gutierrez[1,2,3,4,5], Isabelle Laforest-Lapointe[1,2,3,4,5,7], Nicholas G. Jendzjowsky [1,5,8], Jean-Baptiste Cavin[1,4,8], Fernando A. Vicentini[1,4,8], Catherine M. Keenan[1,4,8], Hena R. Ramay[3], Jumana Samara[1,2,3,4,5], Wallace K. MacNaughton[1,4], Richard J. A. Wilson [1,5,8], Margaret M. Kelly[1,4,9], Kathy D. McCoy [1,3,4], Keith A. Sharkey [1,4,8] & Marie-Claire Arrieta [1,2,3,4,5 ✉]

The gut microbiome consists of a multi-kingdom microbial community. Whilst the role of bacteria as causal contributors governing host physiological development is well established, the role of fungi remains to be determined. Here, we use germ-free mice colonized with defined species of bacteria, fungi, or both to differentiate the causal role of fungi on microbiome assembly, immune development, susceptibility to colitis, and airway inflammation. Fungal colonization promotes major shifts in bacterial microbiome ecology, and has an independent effect on innate and adaptive immune development in young mice. While exclusive fungal colonization is insufficient to elicit overt dextran sulfate sodium-induced colitis, bacterial and fungal co-colonization increase colonic inflammation. Ovalbumin-induced airway inflammation reveals that bacterial, but not fungal colonization is necessary to decrease airway inflammation, yet fungi selectively promotes macrophage infiltration in the airway. Together, our findings demonstrate a causal role for fungi in microbial ecology and host immune functionality, and therefore prompt the inclusion of fungi in therapeutic approaches aimed at modulating early life microbiomes.

[1] Department of Physiology and Pharmacology, University of Calgary, Calgary, AB, Canada. [2] Department of Pediatrics, University of Calgary, Calgary, AB, Canada. [3] International Microbiome Centre, University of Calgary, Calgary, AB, Canada. [4] Snyder Institute for Chronic Diseases, University of Calgary, Calgary, AB, Canada. [5] Alberta Children's Hospital Research Institute, University of Calgary, Calgary, AB, Canada. [6] Department of Clinical Medicine, UiT The Arctic University of Norway, Tromsø, Norway. [7] Department of Biology, Université de Sherbrooke, Sherbrooke, QC, Canada. [8] Hotchkiss Brain Institute, University of Calgary, Calgary, AB, Canada. [9] Department of Pathology and Laboratory Medicine, University of Calgary, Calgary, AB, Canada. ✉email: marie.arrieta@ucalgary.ca

The mammalian intestinal microbiome constitutes a complex community of prokaryotic and eukaryotic microorganisms across high-level clades of the tree of life[1–3]. Structural and metabolic signals derived from this microbial ecosystem engage in crosstalk that helps define the trajectory toward host health or disease, especially early in life[4–7]. The early stages of gut microbiome establishment are more dynamic when compared with adult microbiomes, and environmental factors can influence the composition and diversity of this community, as well as the trajectory of its assembly[7]. Prospective population-level studies have associated microbiome alterations with dysregulation of host physiology and increased susceptibility to immune, metabolic, neurological, and psychiatric diseases[8,9]. However, such studies cannot establish causal relationships nor identify disease-modulating components of the microbiome. These studies have been mainly focused on bacteria, without accounting for the multi-kingdom and multitrophic nature of this microbial ecosystem[10].

Fungi are an integral part of a wide variety of microbial environments[11], including the human gut[12–14], yet their functional ecological role in mammalian gut microbiomes is not well understood. Comparably to bacteria, mother-to-infant and environmental transfer of fungi occur perinatally, resulting in the rapid colonization of different body sites[14]. Gut bacterial and fungal communities inhabit similar intestinal habitats[15], and fungal-bacterial correlations have been identified in human cohort studies[12,13], suggesting integral ecological interactions during early stages of colonization[7].

Gut fungi can confer health benefits to humans. For example, the yeast *Saccharomyces boulardii* is broadly used as an effective probiotic to prevent and treat pathogenic bacterial infections and intestinal complications[16]. However, since the majority of microbiome studies have been centered on bacteria, fungal contribution to host development remains poorly understood. While some research efforts have begun to consider the fungal microbiome (mycobiome), these have mainly focused on its role in human diseases[17,18]. Work from Sokol et al.[19] revealed mycobiome alterations in inflammatory bowel disease (IBD) patients experiencing a flare compared with a healthy cohort or IBD patients in remission. These alterations included an increased fungi/bacteria diversity ratio and an increased abundance of *Candida albicans*, suggestive of fungal overgrowth during inflammation[19]. Another common yeast, *Malassezia restricta*, was identified in the majority of patients carrying the IBD risk allele *CARD9*, a molecule involved in fungal innate immunity[20], possibly implicating this species as one of the factors contributing to IBD.

Early-life gut fungal alterations have also been linked to atopic asthma. Fujimura et al.[13] identified an expansion of *Candida* sp. and *Rhodotorula* sp. in stool samples from US infants that later developed atopy. Similarly, we detected mycobiome alterations in stool of rural Ecuadorian infants that developed atopic-wheeze at 5 years[12]. Differences in the fungal community were more strongly associated with asthma risk than bacterial dysbiosis in the Ecuadorian study, in which we detected an overrepresentation of total fungal sequences and an expansion of the yeast *Issatchenkia orientalis* in children who later developed symptoms[12]. While these population-level studies revealed interesting associations between mycobiome alterations in IBD and asthma, the causal role of fungi in these diseases has not been established.

Causal relationships between microbiomes and diseases require appropriately designed experiments in relevant models. A limited number of studies suggest an immunomodulatory role for the mycobiome. Exacerbated allergic lung inflammation resulted from antibiotic-induced overgrowth of *C. albicans* or *C. parapsilosis* in mice[21,22], or from the expansion of filamentous fungi

following antifungal treatment[23]. Antifungal-induced alterations were further found to worsen lung allergic inflammation via intestinal resident CX3CR1+ phagocytic cells[24]. Further, experimental colonization of mice with *M. restricta* resulted in exacerbated dextran sodium sulfate (DSS)-induced colitis, characterized by CARD9-dependent Th1 and Th17 inflammation[20]. While informative, these studies cannot determine the causal contributions of either fungal or bacterial components, nor can they account for interkingdom interactions. This is relevant because it remains unclear if fungi directly influence the host or if the phenotypic effect results indirectly through fungus-induced changes in the bacterial community. Specifically, the experimental approaches used in previous studies (fungal colonization and/or antimicrobial treatment to specific pathogen free; SPF mice) also impact the bacterial microbiome[25,26], which in turn could be the direct driver of the reported immune effects. Thus, the causal role of gut fungi, and if their effect is mediated directly or via the bacterial community, remains undefined.

The goal of this study was to determine the role of colonizing fungi in host-microbiome interrelationships as they relate to microbiome assembly and host immune and physiological development. Such studies are difficult to achieve using animal models with complex or undefined microbiomes, such as SPF mice. To overcome this challenge, we have utilized a gnotobiotic approach to interrogate the capacity of fungi to (i) stably colonize the mouse gut, (ii) alter microbiome ecology and its response to antimicrobials, (iii) impact gut physiology and systemic immunity, and (iv) influence host susceptibility to allergen and chemically-induced inflammatory diseases. We explored this in germ-free (GF) mice colonized with defined consortia of either bacteria, fungi, or both. Our work shows that intestinal colonization with a consortium of fungal species elicited strong microbiome and immunological shifts that modulated subsequent susceptibility to mucosal inflammation in the distal gut and the lungs.

## Results

**Human-associated fungal species colonize mouse intestinal tract**. We first evaluated the ability of common yeast species to colonize and persist in the mouse gut. Adult GF dams were colonized with defined microbial consortia of either bacteria (12 species; B), yeast (5 species; Y), a mixture of all the bacteria and yeasts (BY), or kept germ-free (GF; Fig. 1a, Supplementary Table 1; see Methods). For bacteria, we utilized the Oligo-MM[12] consortium, which consists of 12 mouse-derived bacterial species that are persistent, inheritable, and eliciting of immune responses in mice that are similar to a complex microbiota[27,28]. For fungi, we selected five yeast species from taxa that commonly colonize the human gut and have been previously linked to atopy and asthma risk early in life[12,13]. Dams were bred, and offspring were further colonized by spreading the dams' abdominal area, including the nipple area, with the same inocula during the first week of life (Fig. 1a). We determined intestinal colonization and GF status by performing quantitative polymerase chain reactions (qPCR) in feces of the offspring (F1 mice), by using bacterial 16S and fungal 18S ribosomal RNA (rRNA) gene-specific primers (Supplementary Table 1), and conventional culture. Bacteria were only detected in B and BY mice, while fungi were detected in Y and BY groups (Fig. 1b–e). Fungal 18S RNA target DNA was two orders of magnitude lower than bacterial DNA at 4 weeks of age (Fig. 1b, c). At 9–10 weeks, fungal DNA was tenfold lower than bacterial DNA in the Y mice, and at this time point the 18S signal was lost in the BY group (Fig. 1c). Given the known limitation of qPCR techniques to amplify low amplicon concentrations, we also cultured feces collected from all mice at 7 weeks of age.

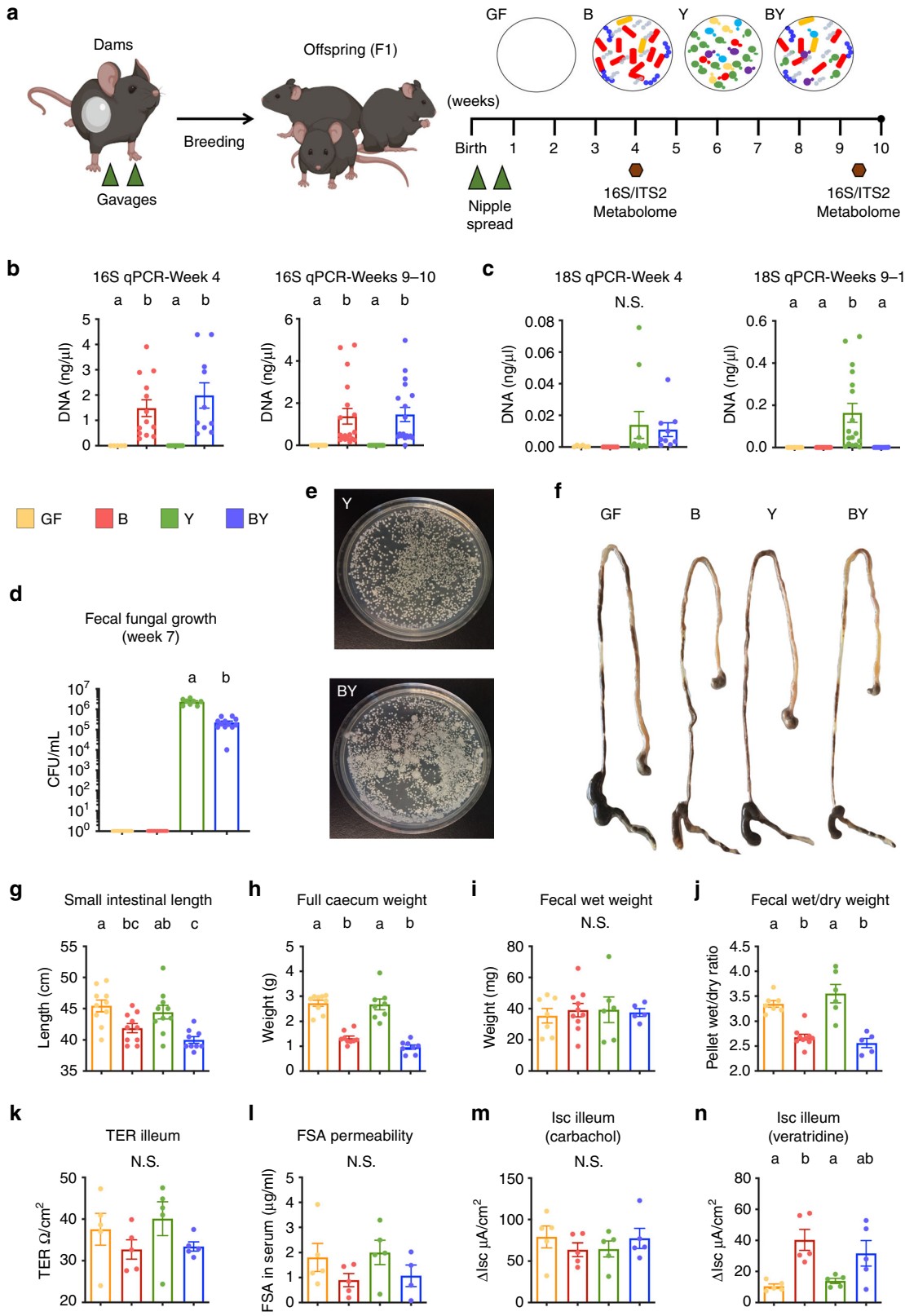

Fungal colonies were obtained from fecal cultures from Y and BY mice only, with fungal cell counts tenfold lower in the BY group ($10^6$ vs. $10^5$ CFU/ml; Fig. 1d, e), confirming that while fungi are able to colonize the mouse gut, bacteria impose an ecological advantage in this model that reduces fungal fitness in this environment.

**Bacteria, but not fungal colonization modulated gut physiology.** We carried out comprehensive gut physiological assessments to determine if colonizing fungi elicit functional changes to the organ harboring them. Anatomical measurements revealed that bacteria, but not fungi, led to a reduction in small intestinal length and full cecum weight, with Y mice exhibiting features

**Fig. 1 Bacterial colonization is essential to induce intestinal anatomical and functional changes of colonized mice. a** Experimental layout for gnotobiotic study. Germ-free dams were orally gavaged twice with a consortium of 12 bacteria (B), 5 yeasts (Y), a combination of both (BY), or kept germ-free (GF). F1 mice were further colonized during the first week of life (see Methods and Supplementary Table 1). Fecal samples (hexagons) were obtained at 4 and 9–10 weeks of age for microbiota quantification (qPCR), taxonomic analysis (16S and ITS2 sequencing), and functional characterization (metabolome). qPCR quantification (standard curve method) of **b** bacterial and **c** fungal DNA in fecal samples by amplification of the 16S and 18S rRNA genes, respectively. **d** Fecal fungal colony counts in YM agar media supplemented with antibiotics (gentamycin + chloramphenicol). **e** Fecal fungal growth in selective medium from representative Y and BY mice. **f** Representation of gross anatomic changes of dissected gastrointestinal tracts across groups (stomach to distal colon). **g** Small intestine length and **h** full cecum weight of gnotobiotic dams at around 20 (17–25) weeks of age. **i** Water content in fecal samples measured by total fecal weight and **j** wet/dry ratio in dams. Gut barrier function measured by **k** ileal transepithelial resistance (TER) and **l** clearance of fluorescein-5(6)-sulfonic acid (FSA) over 4 h. Ileal short-circuit current (ΔIsc) upon stimulation with **m** epithelial stimulator carbachol or **n** neurostimulator veratridine. Data expressed as mean ± S.E.M. **b–d**, **g–j** Color denotes colonization treatment (GF = yellow, B = red, Y = green, BY = royal blue); **b**, **c** Data combined from two different experiments, Week 4: $N_{GF} = 5$, $N_B = 13$, $N_Y = 13$, $N_{BY} = 11$; Week 9: $N_{GF} = 13$, $N_B = 18$, $N_Y = 19$, $N_{BY} = 19$; **d** $N_{GF} = 5$, $N_B = 6$, $N_Y = 8$, $N_{BY} = 12$; **g** $N_{GF} = 10$, $N_B = 10$, $N_Y = 10$, $N_{BY} = 9$; **h** $N_{GF} = 10$, $N_B = 9$, $N_Y = 8$, $N_{BY} = 8$; **i**, **j** $N_{GF} = 7$, $N_B = 10$, $N_Y = 6$, $N_{BY} = 5$; **k**, **m**, **n** $N_{GF} = 5$, $N_B = 5$, $N_Y = 5$, $N_{BY} = 5$; **l** $N_{GF} = 5$, $N_B = 5$, $N_Y = 5$, $N_{BY} = 4$; different letters above bars indicate statistically significant differences defined by one-way ANOVA and Tukey post hoc tests (**a–c**, **g–j**) or two-sided $t$-test (**d**); $P < 0.05$; N.S. no significant differences. Source data are provided as a Source data file.

undistinguishable from GF mice (Fig. 1f–h). Colon length and empty cecum weight remained unchanged across all groups (Supplementary Fig. 1a, b). Bacteria were further essential to induce changes in water absorption and intestinal permeability. Mice from groups B and BY, but not Y, although expressing equitable fecal pellet weight, displayed a reduction of fluid content in feces compared with GF (fecal wet/dry ratio; Fig. 1i, j). Further, examination of the clearance of fluorescein-5(6)-sulfonic acid (FSA), a measurement of intestinal barrier integrity, and ileal transepithelial electrical resistance (TER), a measure of ileal paracellular permeability, showed suggestive reductions in the mice colonized with bacteria (B and BY; Fig. 1k, l), but no significant differences between groups. We then evaluated the ability of the ileum to transport ions by measuring changes in short-circuit current (ΔIsc) after activating enteric nerves with veratridine or directly stimulating the epithelium with carbachol (Fig. 1m, n). Microbial colonization did not influence ΔIsc in response to carbachol; however, GF and Y mice were almost irresponsive to veratridine (low ΔIsc), indicating that neurally-induced ion transport in the ileum relies on bacterial, but not fungal colonization. In contrast to functional changes in the ileal mucosa, no physiological differences were detected in the colon (Supplementary Fig. 1c–f). Altogether, these findings suggest the gut as an organ adapted mainly to bacterial colonization, even if fungi can form stable populations in this environment. Nevertheless, further experiments are needed to confirm if this is a universal response to bacteria and/or fungi, or if exclusive to the microbial consortia chosen.

**Bacterial-fungal co-colonization exacerbated DSS-induced colitis.** Fungal dysbiosis has been associated with IBD severity[19,20], although little is known about the specific contribution of common fungal species to this group of disorders. To assess if fungal colonization directly impacts the development of colitis, 7-week-old gnotobiotic mice colonized with the aforementioned consortia were treated with 1.5% DSS for 5 days (Fig. 2a). Except for one animal in the BY group, all mice survived 5 days of DSS treatment. As compared with mice colonized with bacteria (B and BY), exclusive yeast colonization resulted in decreased inflammatory features, comparable to those of DSS-untreated mice (naive) and GF mice (Fig. 2b–e), indicating that these yeasts alone are insufficient to trigger the immune mechanisms necessary for overt DSS colitis. Nevertheless, when co-colonized with bacteria, BY mice exhibited the highest colitis severity, as measured by body weight loss, spleen weight, onset of positive blood in stool, and stool lipocalin-2 (Lcn-2, a marker of neutrophil infiltration)[29] (Fig. 2b–d). While BY mice exhibited

worse disease scores, DSS-treated colon tissue of B mice displayed greater concentrations of pro-inflammatory cytokines IFN-γ, TNF-α, and IL-22 (Fig. 2e), and Y mice displayed cytokine patterns similar to GF mice (Fig. 2e). Histopathological scoring of DSS-treated colons showed an increase in goblet cell depletion in B and BY groups compared with GF and naive animals (Supplementary Fig. 2b–c) yet no overall differences in total inflammation score for the different colonization groups (Supplementary Fig. 2a). These results suggest that while DSS colitis relies on bacterial signals, fungal co-colonization can modulate the severity and immunophenotype of the immune response.

**Fungi induced strong ecological changes to the gut microbiome.** We sequenced the 16S rRNA (bacteria) and ITS2 region of the nuclear ribosomal internal transcribed spacer (ITS) genetic marker (fungi) to study the impact of fungal colonization on the bacterial microbiome, and to detect interkingdom interactions. We compared bacterial and fungal diversity and composition at 4 and 9–10 weeks to evaluate microbiome shifts during early life and adulthood, respectively. To further unravel bacterial-fungal interactions, we examined the bacterial and fungal microbiome response to community disruption, by treating two additional mouse groups with the antibiotic Augmentin® (amoxicillin-clavulanate; BY + Abx) or the antifungal Fluconazole® (BY + Afx) during the second week of life (see Methods). We first evaluated the influence of the type of colonization (treatment) and mouse age on bacterial and fungal beta-diversity using permutational multivariate analysis of variance (PERMANOVA)[30] on Bray–Curtis dissimilarities among samples (Supplementary Tables 2 and 3), while carefully controlling for cage effects. We observed that interkingdom interactions have a strong effect on the early-life assembly on the bacterial and fungal communities. Colonization schemes explained 15.14% and 38.97%, whereas mouse age explained 14.08% and 3.28% of the variance in the bacterial and fungal community structures, respectively ($P < 0.001$, PERMANOVA; Supplementary Tables 2 and 3, Fig. 3a, b). When the microbial communities were analyzed per time point, treatment explained significant shifts in bacterial and fungal beta-diversity at 4 weeks (62.91% and 74.25% of variance, respectively, $P < 0.001$, PERMANOVA; Supplementary Tables 2 and 3). At 9 weeks of age, the strongest driver of bacterial beta-diversity was the cage effect (27.84% of variation, $P = 0.005$, PERMANOVA; Supplementary Table 2). In comparison, the fungal community exhibited longer-lasting variations in beta-diversity due to bacterial colonization or antimicrobial disturbances (74.25% of variance, $P < 0.001$, PERMANOVA; Supplementary Tables 3, Fig. 3a, b),

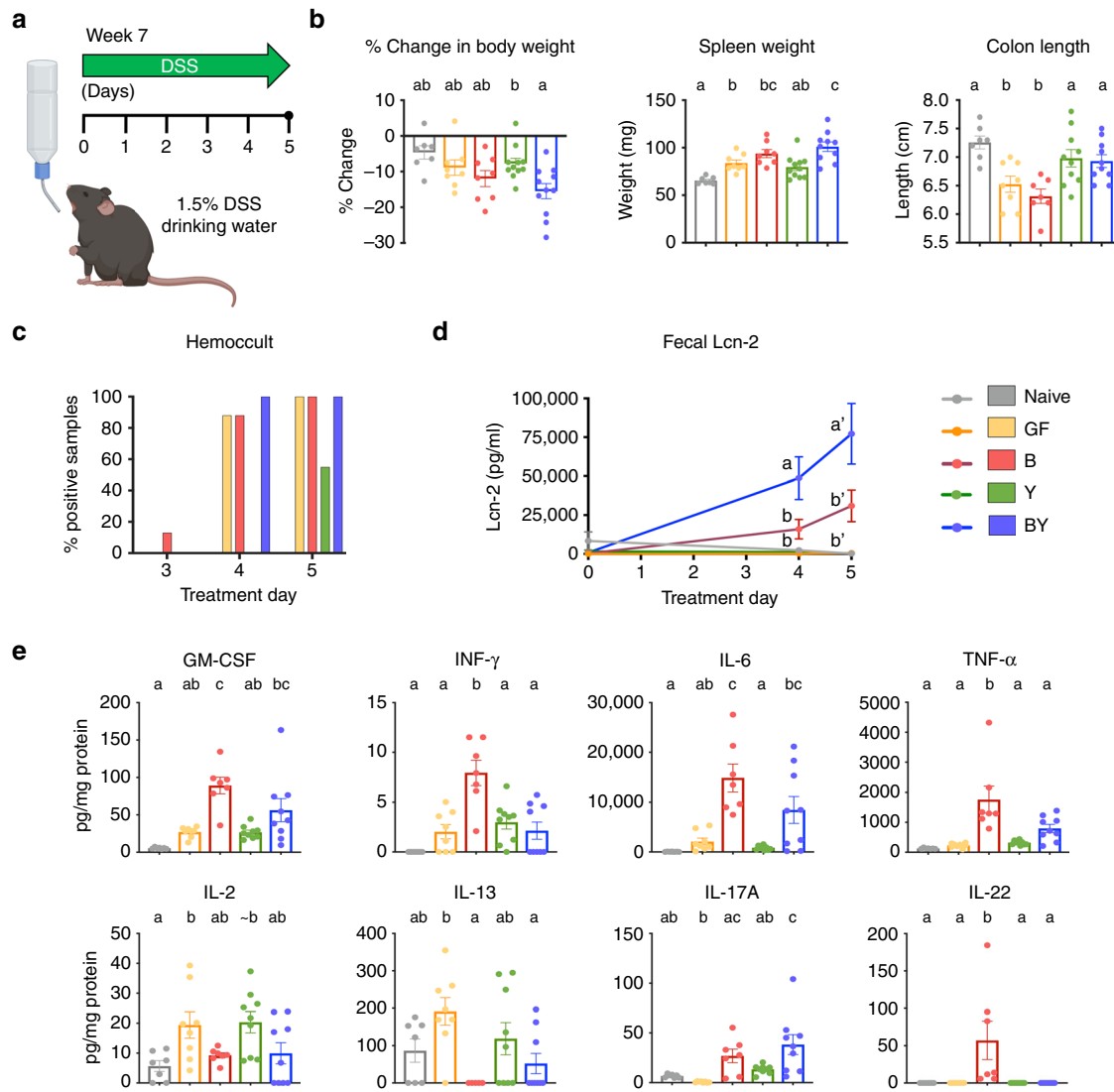

**Fig. 2 Fungal immune modulation of DSS-induced colitis. a** Experimental layout for DSS study. Seven-week-old gnotobiotic F1 mice were treated with 1.5% DSS for 5 days and colonic inflammation was assessed at end of treatment. A group of colonized animals were not treated with DSS and denoted naive (see Methods). **b** Percentage weight change between days 1 and 5, spleen weight, and colon length after 5 days of DSS treatment. Measurements of **c** stool occult blood and **d** stool Lcn-2 throughout DSS treatment. **e** Pro-inflammatory cytokines detected in inflamed colon lysates by electrochemilumine-scence (MSD). **b–e** Data expressed as mean ± S.E.M.; color denotes colonization and DSS treatment (Naive = gray, GF = yellow, B = red, Y = green, BY = royal blue); $N_{NAIVE} = 7$, $N_{GF} = 8$, $N_B = 8$, $N_Y = 11$, $N_{BY} = 11$; different letters above bars indicate statistically significant differences defined by one-way ANOVA and Tukey post hoc tests; $P < 0.05$. Source data are provided as a Source data file.

highlighting that the fungal community is less resilient to early-life alterations.

We also compared species alpha-diversity (Shannon diversity index) across treatments (non-parametric Kruskal–Wallis tests followed by post hoc Dunn tests with Benjamini–Hochberg false discovery rate, FDR correction). For bacteria, no differences across treatment groups were detected for either time point (Fig. 3c). In contrast, fungal alpha-diversity was drastically reduced in mice harboring only fungi when compared with BY mice at 4 weeks of age. This reduction was comparable to the effect of antimicrobials (Fig. 3d), indicating that gut fungal diversity benefits from bacteria during early-life colonization. Fungal alpha-diversity increased by week 9 in all groups except in the community challenged with the antifungal (Fig. 3d). Alto-gether, these results show that bacterial and fungal population reciprocally change each other, with stronger and longer-lasting effects of bacteria on the fungal community. These interkingdom

effects were comparable to the effects of antimicrobials, pointing to fungal colonization as an important factor driving microbiome assembly.

To identify the microbial taxa driving these community shifts we determined the differential relative abundance of the microbial species across treatment groups (non-parametric Kruskal–Wallis tests followed by post hoc Dunn tests with Benjamini–Hochberg FDR correction). For the bacterial microbiome, fungal coloniza-tion impacted less predominant taxa, including *Bifidobacterium longum* and *Lactobacillus reuteri*. Fungal colonization distinctly antagonized the growth of *L. reuteri*, which was drastically decreased in BY mice compared with B or B + Afx mice (Supplementary Fig. 3a).

We detected stronger compositional changes for all fungal spe-cies (Fig. 3e). The top two colonizers, *C. albicans* and *Rhodotorula mucilaginosa* represented over 90% of the fungal communities and exhibited the largest shifts across treatments. *C. albicans*

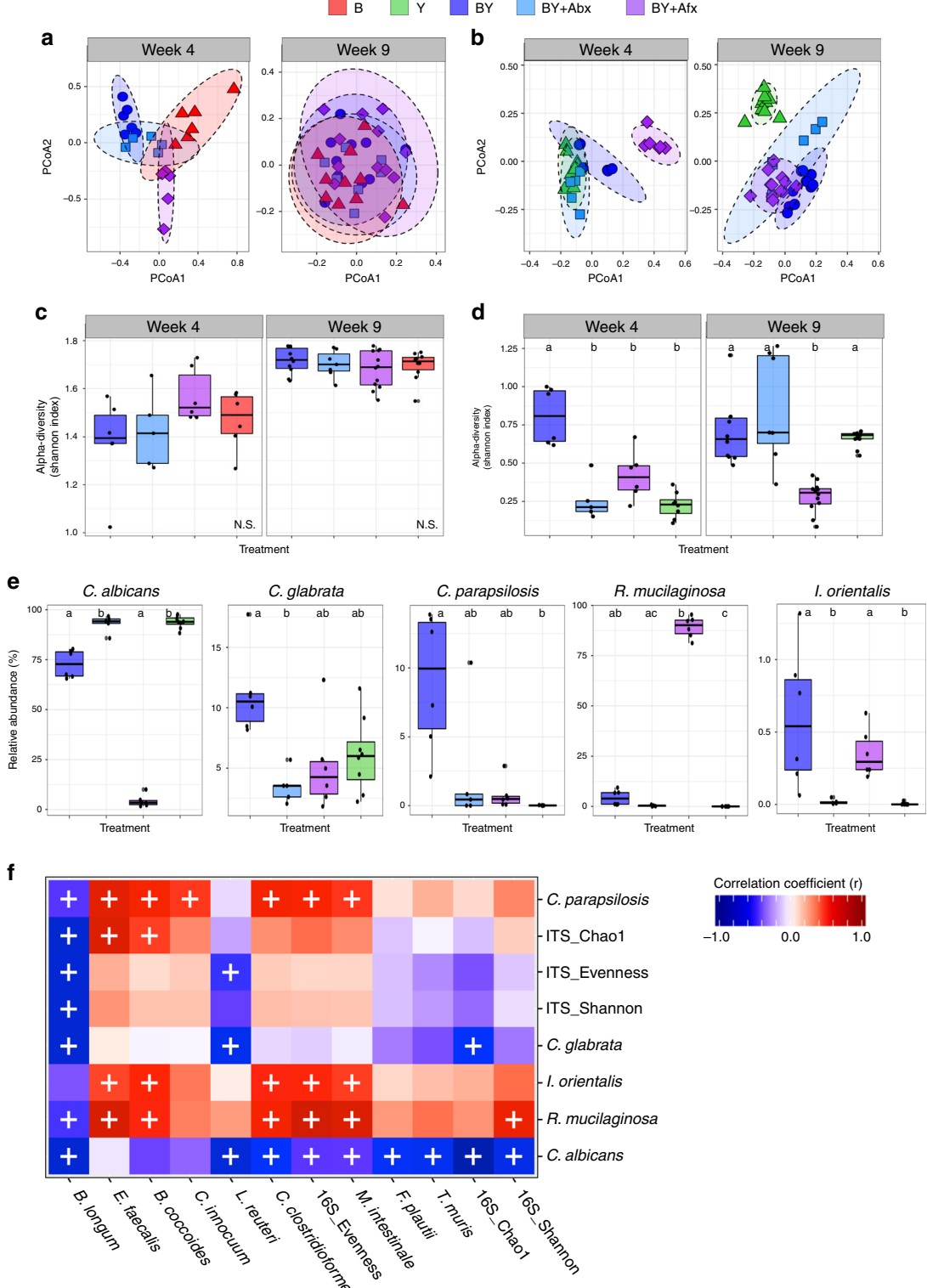

**Fig. 3 Fungal colonization and antimicrobial treatments influence gut microbiome ecology.** Ecological community analyses of 16S and ITS2 sequences. PCA ordination of variation of **a** bacterial or **b** fungal beta-diversity of mice gut microbial communities based on Bray–Curtis dissimilarities among samples across treatments and experimental time points. Plots of **c** bacterial or **d** fungal alpha-diversity (Shannon index) across groups. **e** Relative abundances of fungal species at 4 weeks. **a–e** Color denotes colonization treatment (B = red, Y = green, BY = royal blue, BY + Abx = cyan blue, BY + Afx = purple); Week 4: $N_B = 6$, $N_Y = 8$, $N_{BY} = 6$, $N_{Abx} = 5$, $N_{Afx} = 6$; Week 9: $N_B = 10$, $N_Y = 11$, $N_{BY} = 10$, $N_{Abx} = 7$, $N_{Afx} = 12$; **c–e** Boxplots indicate median (inside line) and 25th and 75th percentile as the lower and upper hinges, respectively; different letters above bars indicate statistically significant differences defined by Kruskal–Wallis with post hoc Dunn tests and FDR corrected; $P < 0.05$; N.S. no significant differences. **f** Heat map of biweight correlations between bacterial (y-axis) and fungal species (x-axis) in feces collected at 4 weeks of age. Color denotes positive (red) and negative (blue) correlation values. Significant correlations are denoted with a cross and defined by the BiCOR method with FDR correction; $P < 0.05$. Source data are provided as a Source data file.

dominated the community in all groups, except in mice treated with fluconazole, in which *R. mucilaginosa* was the dominant taxon (Fig. 3e), suggesting interspecies competition or resistance to fluconazole in vivo (all fungal strains were susceptible to fluconazole in vitro). This result was also quantified by species-specific qPCR, confirming that these shifts in relative abundance reflected real changes in cell concentration (Supplementary Fig. 3b, c). Co-colonization with bacteria resulted in increased abundance of *C. parapsilosis* and *I. orientalis*, whereas *C. albicans* dominated in the absence of bacteria or in antibiotic-treated mice (Fig. 3e). At week 9, smaller differences remained for *L. reuteri*, *Enterococcus faecalis*, *Clostridium innocuum*, and *B. longum*, while major compositional changes involving all fungal species remained across treatment groups (Supplementary Fig. 3d, e).

To gain insight into the inter-domain ecological interactions at play in these defined communities, we carried out correlation analysis between the 16S and ITS2 datasets, using the relative abundances of the Amplicon Sequence Variants (ASVs) that explained >98% of the fungal and bacterial communities, merged by species name (Biconjugate A-Orthogonal Residual, BiCOR method with FDR correction). We identified strong interkingdom correlations in line with the changes in relative abundance across treatment groups. The presence of *C. albicans* was inversely correlated with six different bacterial species (*L. reuteri*, *Muribaculum intestinale*, *Flavonifractor plautii*, *Turicimonas muris*, *B. longum*, and *C. clostridioforme*; Fig. 3f). Among these, bacterial species *L. reuteri* and *B. longum* were also negatively correlated with *C. glabrata*. As was previously observed (Supplementary Fig. 3a), *B. longum* was the bacterial species most negatively impacted by the presence of fungi, showing inverse correlations with all *Candida* species, as well as *R. mucilaginosa* (Fig. 3f). This analysis also revealed positive correlations between fungal species *I. orientalis*, *C. parapsilosis*, and *R. mucilaginosa* and the bacteria *C. clostridioforme*, *B. coccoides*, *E. faecalis*, and *M. intestinale* (Fig. 3f). Interestingly, the abundance of certain species impacted alpha-diversity measures of the opposite domain. For example, *B. longum* was negatively correlated with fungal alpha-diversity (Shannon index), richness (Chao1), and evenness (Shannon/logChao1), whereas *C. albicans* was inversely associated with bacterial alpha-diversity and richness (Fig. 3f). These findings convincingly show that fitness of some of the most common yeast species in the mouse gut benefit from the presence of bacteria, while the growth of some bacterial strains, such as *B. longum* and *L. reuteri*, is strongly antagonized by fungal colonization.

**Intestinal fungi contributed minimally to the fecal metabolome.** Microbial metabolites produced in the gut are predicted to modulate immune development, though very little is known about the contribution of fungal species. We explored the functional changes attributed to fungal colonization by determining the fecal metabolomes for each of our gnotobiotic groups by liquid chromatography coupled to mass spectrometry (LC-MS). Principal component analysis (PCA) performed on the relative amounts of 150 metabolites, which were obtained from feces collected at weeks 4 and 9, demonstrated that the metabolome profile of Y mice was similar to GF, which were both very different from the groups colonized with bacteria, B, BY, BY + Abx, and BY + Afx (Fig. 4a). Statistical analysis revealed 99 metabolites with significantly different amounts between the groups ($P < 0.05$, ANOVA with FDR correction; Supplementary Table 4). The heat map of the top 25 most differentially detected metabolites shows clustering of Y and GF samples separately from bacteria-colonized groups (Fig. 4b). Metabolites detected at lower levels in the GF and Y groups were associated with the metabolism of

butanoate, ubiquinone, pantothenate, coenzyme A, and mostly aromatic amino acids (Supplementary Table 4). The presence of bacteria reduced levels of several oligo and monosaccharides and a large number of amino acids, hinting at bacterial consumption of these compounds (Supplementary Table 4). These results thus suggest that fungal metabolites do not contribute significantly to overall fecal metabolome of colonized mice.

Although the metabolome profiles of the Y and GF groups appeared very similar, we identified 22 metabolites with significantly different levels between Y and GF samples at 9 weeks ($P < 0.05$, T-test with FDR correction; Fig. 4c, Supplementary Table 5). The Y group showed increased levels of metabolites produced in the citric acid cycle and butanoate metabolism, while GF mice had significantly elevated levels of metabolites associated with metabolism of histidine, asparagine, serine, and nitrogen (Supplementary Table 5). The most significant changes originating from fungal colonization were detected for nicotinic acid, 3-hydroxybutyric acid, fumaric acid, and L-asparagine (Fig. 4d, Supplementary Table 5). This confirmed that the colonizing fungi are metabolically active in the murine intestinal tract.

We also looked at metabolome profiles of the small intestine content for B, BY, Y, and GF mice at week 4 (Supplementary Fig. 4). As opposed to the fecal metabolome data, we did not observe any specific grouping of the mice based on the metabolome of the small intestinal luminal content, neither by PCA of 124 quantified metabolites (Supplementary Fig. 4a) nor hierarchical clustering analysis of the relative abundances for the top 25 detected metabolites (Supplementary Fig. 4b). This observation agrees with the fact that highest bacterial load is located in the large intestine, with the small intestine harboring generally lower bacterial amounts[31]. In conclusion, although fungal colonization with this defined consortium promotes strong alterations in the bacterial microbial community, its impact on the fecal metabolome profile of colonized mice is limited.

**Fungal colonization altered early-life systemic and gut immunity.** To examine the immune consequences of fungal colonization, we characterized the immune response at 4 weeks of age by using flow cytometry of extracellular and intracellular markers in unstimulated splenocytes (flow cytometry gating strategy outlined in Supplementary Fig. 5), immunoglobulin (Ig) quantification in serum, and multi-array cytokine determination in jejunum and colon lysates. Fungal colonization with this consortium promoted broad systemic immunological changes to immune cell populations and their secretion of cytokines in the spleen (Fig. 5a, b, Supplementary Fig. 6a, b). BY mice exhibited shifts in CD19$^+$ B cells, CD3$^+$ T cells, FoxP3$^+$ regulatory T cells (Tregs), and Lin$^-$ (CD19$^-$CD3$^-$) myeloid cells, compared with B and GF groups (Fig. 5a). Antibiotic treatment resulted in similar changes in B and T-cell populations, compared with B and GF mice (Fig. 5a). Within the Lin$^-$ population we also detected alterations in macrophage subpopulations. BY-colonization increased the proportion of F4/80$^+$MHC-II$^+$CD11c$^-$ macrophages (Fig. 5a). Changes in Tregs and macrophage subpopulations are likely attributed to fungi as Y and BY mice exhibited the same patterns, compared with GF and B mice (Fig. 5a). Antimicrobial treatments to BY mice resulted in changes in most of the splenocyte populations, compared with untreated BY mice (Fig. 5a, Supplementary Fig. 6a). Systemic immune changes were also evident from cytokine secretion by splenic cells and serum antibodies. Percentages of splenocytes producing IL-4, IL-6, IL-10, and IL-12 cytokines in BY groups were significantly increased from GF or B (Fig. 5b), suggesting a synergistic immune effect when a host is colonized with fungi and bacteria. Increased IL-4 and IL-6

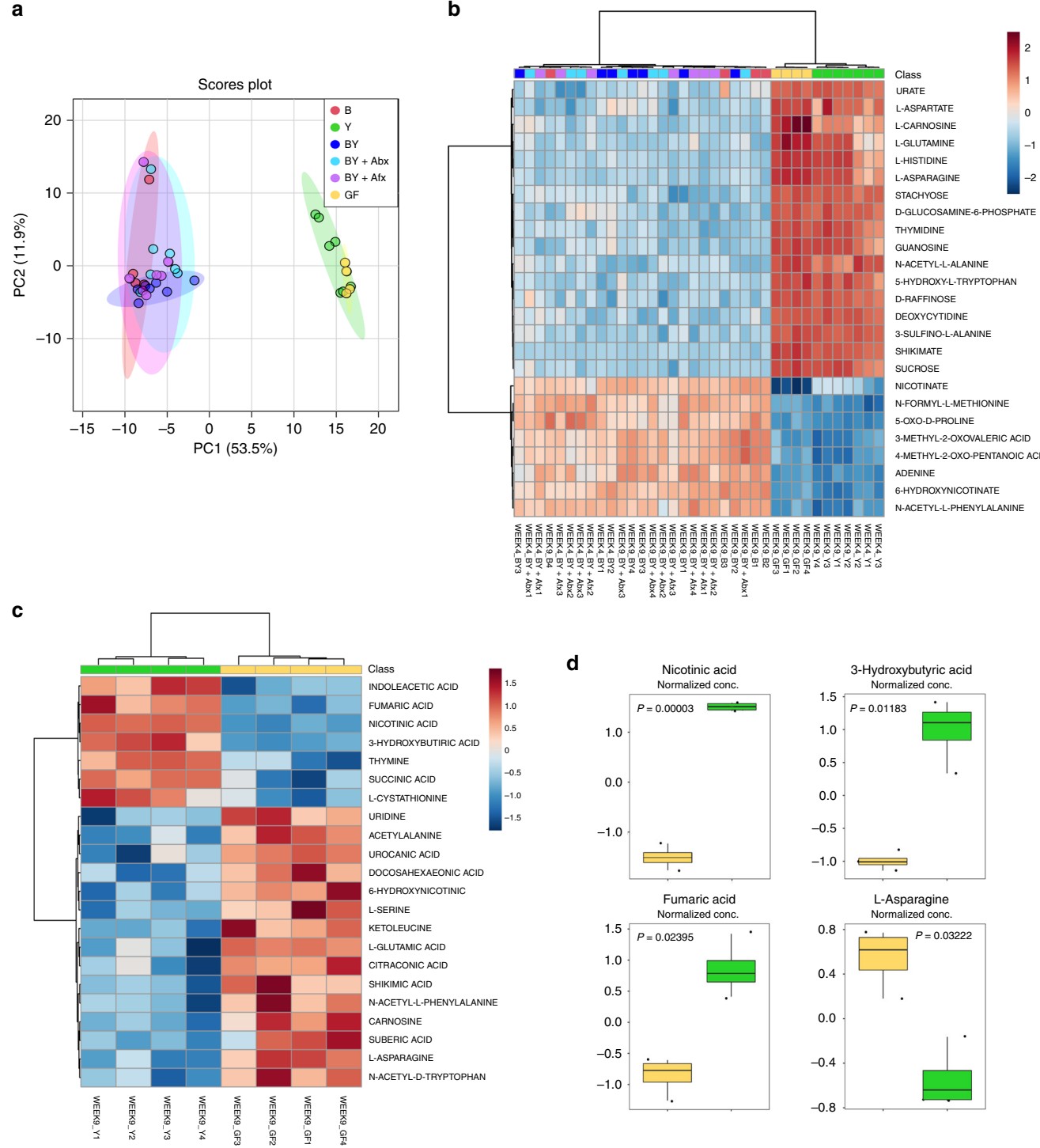

**Fig. 4 Bacteria colonization is the main driver of changes in the fecal metabolome. a** Principal component analysis score plot of 150 metabolites detected in fecal samples of gnotobiotic mice at 4 and 9 weeks of age. **b** Heat map of top 25 differentially expressed metabolites detected in 4- and 9-week fecal samples. **c** Heat map of 22 metabolites differentially expressed between GF and Y groups at 9 weeks. **d** Strongest metabolic differences detected between GF and Y groups at 9 weeks. **a–d** Color denotes colonization treatment (GF = yellow, B = red, Y = green, BY = royal blue, BY + Abx = cyan blue, BY + Afx = purple); $N_{week4}$ = 3 per group; $N_{week9}$ = 4 per group. **a, b** Statistically significant differences between groups determined by one-way ANOVA with Fisher's post hoc FDR correction; $P < 0.05$ (refer to Supplementary Table 4). **c, d** Statistically significant differences defined by two-tailed $t$-test with FDR correction; $P < 0.05$ (refer to Supplementary Table 5). **d** Boxplots indicate median (inside line) and 25th and 75th percentile as the lower and upper hinges, respectively. Source data are provided as a Source data file.

secretion was mainly driven by CD4$^+$ T and Lin$^-$CD11c$^+$ cells, while Lin$^-$CD11b$^+$ cells displayed increased expression of IL-6 and IL-12 (Supplementary Fig. 6b). Exclusive fungal colonization resulted in decreased serum IgG3, as well as increased IgG1

(Fig. 5c), and a non-significant increase in IgE compared with B group (Supplementary Fig. 6c). As well, BY and BY + Abx mice displayed increased levels of IgA and IgG3 (Fig. 5c). Importantly, these striking systemic immune changes were not a result of

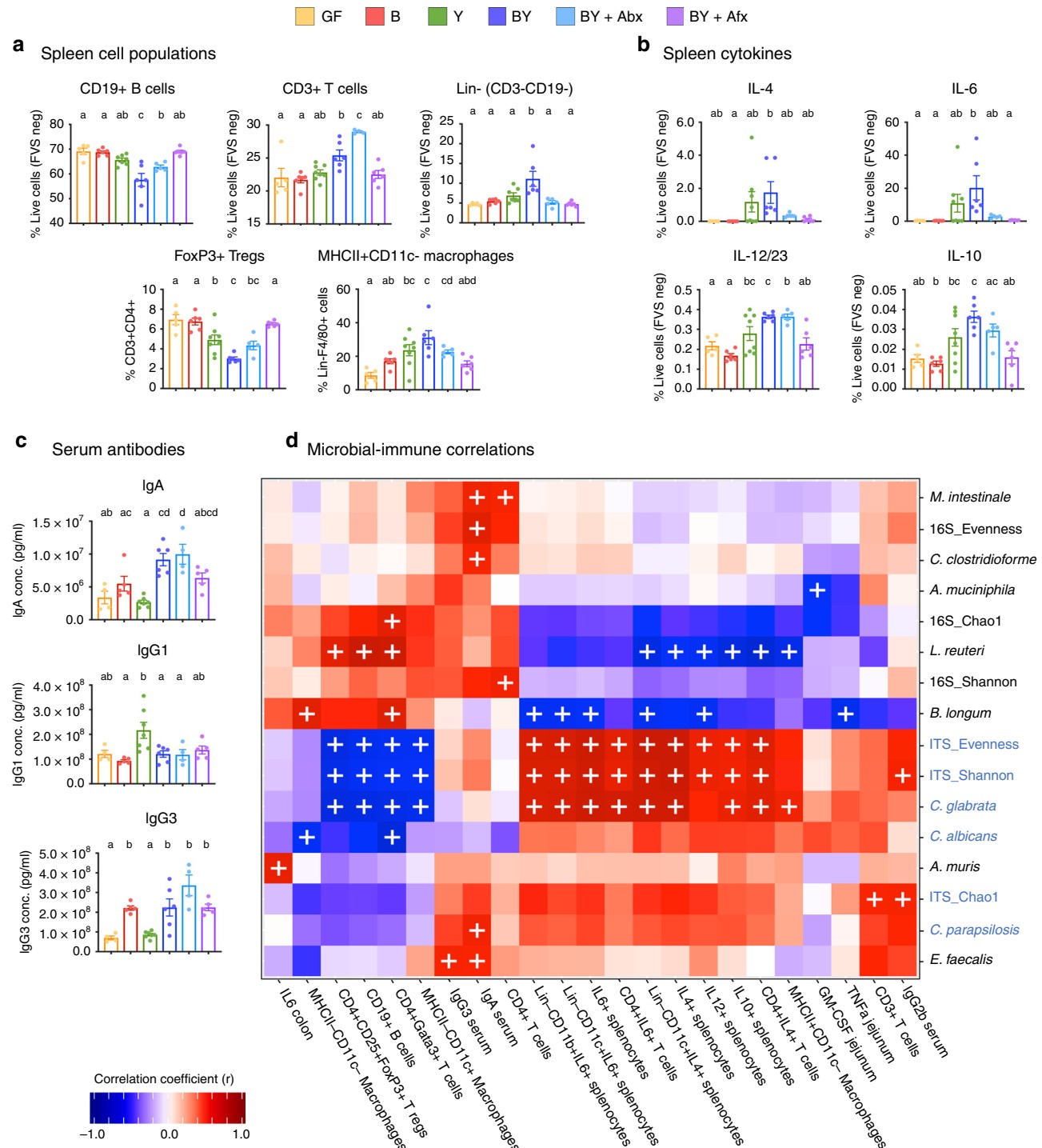

**Fig. 5 Fungal colonization alters early-life systemic immunity in mice.** Percentage of **a** unstimulated splenic cell populations, and **b** cytokine-producing unstimulated splenocytes from 4-week-old gnotobiotic mice. Cells were stained with intra- and extracellular marker-specific antibodies and quantified by flow cytometry. **c** Serum antibody concentrations detected by electrochemiluminescence (MSD). **a–c** Data expressed as mean ± S.E.M. Color denotes colonization treatment (GF = yellow, B = red, Y = green, BY = royal blue, BY + Abx = cyan blue, BY + Afx = purple); $N_{GF} = 5$, $N_B = 6$, $N_Y = 8$, $N_{BY} = 6$, $N_{Abx} = 5$, $N_{Afx} = 6$; different letters above bars indicate statistically significant differences defined by one-way ANOVA and Tukey post hoc tests; $P < 0.05$. **d** Heat map of biweight correlations between merged 16S (black) and ITS (blue) ASVs of the same species (x-axis) and reported immune features (y-axis) at week 4. Color denotes positive (red) and negative (blue) correlation values. Significant correlations are denoted with a cross and defined by the BiCOR method with FDR correction; $P < 0.05$. Source data are provided as a Source data file.

systemic fungal infection as fungal cultures of the kidneys, a classic marker of fungemia[32], were negative in all mice.

Microbial colonization also drove mucosal immune changes in the jejunum and colon (Supplementary Fig. 6d, e). Cytokine levels in jejunum lysates of GF mice consistently showed high levels of pro-inflammatory cytokines, IL-6, IL-17A, IL-21, INF-γ, TNF-α, and GM-CSF (Supplementary Fig. 6d). Microbial colonization decreased these cytokine levels, but only with bacterial colonization (B and BY mice) whereas exclusive colonization with fungi was insufficient to induce this effect, except for IL-21

(Supplementary Fig. 6d). Microbial colonization also affected cytokine level in the colon but fewer changes were attributed to fungal colonization (Supplementary Fig. 6e).

To further characterize the specific microbial drivers of the aforementioned immunological changes, we correlated the relative abundance of bacterial and fungal species and sequencing statistics with the reported immune features (BiCOR method with FDR correction). Remarkably, several of the significant correlations observed were ascribable to fungal colonizers (Fig. 5d). *C. glabrata* promoted the most striking systemic changes observed, including the increased proportions of IL-4, IL-6, and IL-10-producing splenocytes, as well as decreasing proportions of B cells, Tregs, Gata3-producing T cells, and MHC-II$^-$CD11c$^+$ macrophages (Fig. 5d). *C. parapsilosis* was also strongly correlated to an increase in serological levels of IgA (Fig. 5d). Fungal community alpha-diversity and evenness also significantly impacted systemic immunity. Fungal alpha-diversity (Shannon) and evenness were positively correlated to increased proportions of pro-inflammatory cytokine-producing cells, and reduction of B cells, Tregs, Gata3-producing T cells, and MHC-II$^-$CD11c$^+$ macrophages (Fig. 5d). Fungal richness (Chao1) was associated with increased T-cell proportions and serum IgG2b (Fig. 5d).

We detected more correlations between some of the bacterial colonizers and immune markers. *B. longum* and *L. reuteri* were correlated with reduced secretion of IL-4 and IL-12 pro-inflammatory cytokines, and increased proportions of Gata3-producing CD4$^+$ T cells (Fig. 5d). Furtherly, *B. longum* reduced proportion of IL-6-expressing splenocytes, while *L. reuteri* was correlated with reduced secretion of IL-10 cytokine, reduced proportion of MHC-II$^+$CD11c$^-$ macrophages, and increase in Tregs and B cell populations (Fig. 5d). Bacterial community Shannon and Chao1 indexes were also associated to increased proportion of CD4$^+$ T splenocytes and Gata3$^+$ CD4$^+$ T cells, respectively (Fig. 5d).

We also applied sparse generalized canonical correlation analysis (SGCCA) to further determine the relevance of the top 15 colonizers from the 16S and ITS2 datasets on the measured immunological features. Relevance network plots identified *C. albicans* and *R. mucilaginosa* as the two fungal taxa with most relevant correlations with systemic immune cell populations (Supplementary Fig. 7). This method confirmed many of the above-mentioned correlations (Fig. 5d), as well as a negative correlation between *C. albicans* and jejunum IL-2 levels (Supplementary Fig. 7). SGCCA also detected several bacterial species, *C. innocuum*, *C. clostridioforme*, *M. intestinale*, *Blautia coccoides*, and *A. muciniphila* as relevant for splenocyte populations and their cytokine responses (Supplementary Fig. 7). Altogether, these results highlight that fungal colonization is strongly correlated with the observed systemic immunological features.

**Gut fungal colonization modulated OVA-induced airway inflammation**. Fungal alterations are associated with subsequent asthma susceptibility in humans[12,13], yet causality remains to be determined. We explored the impact of fungal colonization on host susceptibility to airway inflammation by challenging mice with ovalbumin (OVA; Fig. 6a, see Methods). We observed that GF mice were highly susceptible to OVA-airway inflammation, demonstrating increased bronchoalveolar lavage (BAL) cellularity and eosinophilia, relative to naive (non-OVA challenged) animals (Fig. 6b), as previously described[33]. Remarkably, compared with naive mice, exclusive fungal colonization also resulted in elevated measures of lung inflammation (Fig. 6b). Co-colonization with yeasts (BY) did not impact overall total cell counts in BAL, compared with the B group, suggesting that bacterial colonization

alone is sufficient to reduce airway inflammation subsequent to OVA challenge (Fig. 6b). However, antimicrobial treatments in BY + Abx and BY + Afx groups led to increased airway inflammation (Fig. 6b), suggesting that community perturbations to the bacterial and fungal communities are necessary and sufficient to increase host susceptibility to allergic airway inflammation.

Differential cellular counts of the inflammatory infiltrate highlighted the distinctive phenotype and tone of the inflammatory responses. Increased eosinophilia was observed in GF, BY + Abx, and BY + Afx groups, while the increased cellular infiltrate in the Y group was marked by increased counts of monocytes/ macrophages (Fig. 6c, d), suggesting that the colonizing fungi can modulate the immune phenotype of airway inflammation. Overall, the presence of bacteria (B and BY) but not fungi reduced acute airway inflammation in this model, suggesting that homeostatic control of lung allergic responses relies on bacterial signals, and that fungal gut colonization can skew the inflammatory response, resulting in increased macrophage infiltration in the bronchoalveolar space during allergic inflammation.

## Discussion

Using a gnotobiotic approach, we tested the direct and indirect causal effects of fungal colonization on intestinal physiology and systemic immune development, as well as their ecological role in the microbial community. To our knowledge, this is the first time the role of fungi has been investigated in the absence of bacteria, enabling the study of their direct contribution to host development. Our work provides a systematic analysis on the role of fungi, showing that while the mouse gut is more physiologically equipped to harbor bacteria, fungi strongly impact microbiome dynamics (Fig. 3, Supplementary Fig. 3), and promote robust local and systemic immunological changes (Fig. 5, Supplementary Figs. 6 and 7) that influence the immune phenotype of intestinal and lung inflammatory responses (Figs. 2 and 6).

Previous work has suggested that fungi are unable to colonize the intestinal tract of healthy humans, proposing that fungal detection in fecal samples originates from fungal transiting through the gastrointestinal tract from food sources or the oral cavity[34]. In contrast, our mouse model consistently showed that, even with periodic cage bedding changes and sterile food sources, we were able to culture fungi from gnotobiotic fecal samples several weeks after initial inoculation (Fig. 1d, e). These results indicate that fungi are gut dwellers, albeit in lower concentrations than gut bacteria. Successful fungal colonization in the absence of bacteria further indicates that the latter are not required for fungi to engraft in the mouse gut. Fungi grew in higher concentration in the absence of bacteria, in line with previous findings showing interkingdom competition or antagonism between bacteria and fungi in other ecosystems, such as the rhizosphere and soil[35]. However, our data also revealed that fungal community alpha-diversity and the relative abundances of several fungal species increased in the presence of bacteria, suggesting synergistic interkingdom relationships, which have also been reported between bacteria and fungi in other ecosystems[36]. Thus, it is likely that the gut microbiome hosts a broad array of ecological interactions between fungi and bacteria.

While not reflective of natural conditions, exclusive fungal colonization is a valuable experimental benchmark to differentiate the effects of bacteria and fungi, and potential synergism or antagonism in these effects. This condition was essential to determine that, although this fungal consortium colonized the mouse gut, their presence was insufficient to induce any physiological or morphological changes in the ex-GF mice. Intestinal anatomy and physiology in mice exclusively colonized with fungi,

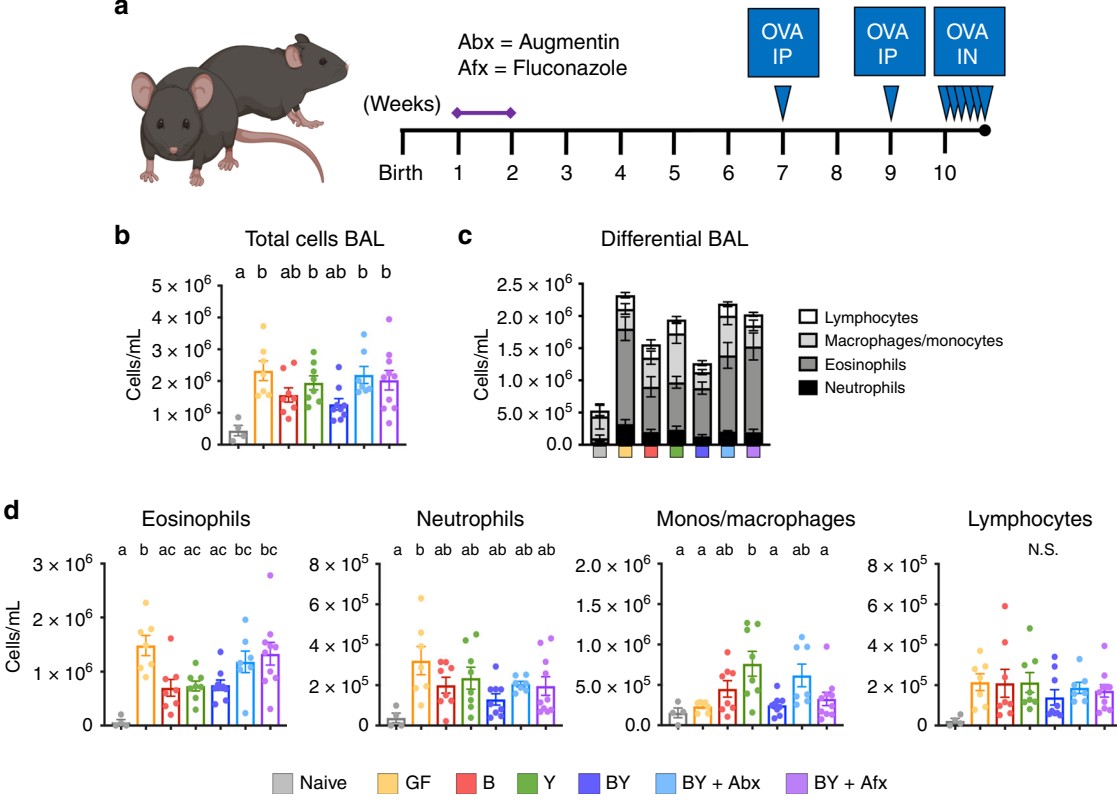

**Fig. 6 Fungal colonization modulates OVA-induced airway inflammation phenotype. a** Experimental layout for OVA-induced airway inflammation. Two groups of BY animals were additionally treated with antimicrobials Augmentin (Abx) or Fluconazole (Afx) during the second week of life. F1 mice were systemically sensitized with two intraperitoneal (IP) injections of OVA+Alum at weeks 7 and 9. Afterward, F1 mice were challenged intranasally (IN) with OVA, for 5 consecutive days at week 10. A group of mice were kept unchallenged and denoted naive (see Methods). **b** Total cellular count in BAL fluid following OVA challenge. **c** Compiled differential counts of cellular infiltrate in gnotobiotic groups. **d** Differential leukocyte counts in challenged mice. **b–d** Data expressed as mean ± S.E.M.; color denotes colonization and OVA challenge (Naive = gray, GF = yellow, B = red, Y = green, BY = royal blue, BY + Abx = cyan blue, BY + Afx = purple); $N_{NAIVE} = 5$, $N_{GF} = 7$, $N_B = 8$, $N_Y = 8$, $N_{BY} = 9$, $N_{Abx} = 7$, $N_{Afx} = 10$; **b, d** different letters above bars indicate statistically significant differences defined by one-way ANOVA and Tukey post hoc tests; $P < 0.05$. Source data are provided as a Source data file.

including the most common human colonizer, *C. albicans*, were essentially identical to the GF mice (Fig. 1, Supplementary Fig. 1), suggesting that the reported adaptations of the gut tissue to microbial colonization[31] may not extend to these fungal organisms. Our proof-of-concept study cannot rule out, however, that other fungal species not included in this consortium may induce a physiological response in the gut.

Our data also bring to light that the gut environment is conducive to very strong interkingdom bidirectional interactions. The presence of fungi impacted bacterial community structure, although bacteria had a stronger and longer-lasting effect on the fungal community (Fig. 3, Supplementary Fig. 3). Co-colonization with bacteria strongly altered the fungal community's beta-diversity, and significantly increased alpha-diversity and richness (Fig. 3, Supplementary Fig. 3). Interestingly, the impact of fungal colonization on bacterial beta-diversity and richness was stronger at 4 weeks of age (Fig. 3), supporting the concept of a window of opportunity early in life, during which the bacterial microbiome is more susceptible to alterations[7]. Fungal colonization additionally shapes microbiome trajectory during early assembly, rendering fungi as a decisive factor in host development during early life through interkingdom mechanisms that remain undefined and important to elucidate.

Microbial colonization early in life is critical for immune education[37,38] and our work provides causal evidence that fungi, in the absence or presence of bacteria, stimulate systemic immune responses in neonates (Fig. 5, Supplementary Fig. 6). Most

systemic immune changes resulted from the synergistic effects of both bacterial and fungal colonization (Fig. 5a, b, Supplementary Fig. 6a, b), including shifts in major immune cell subtypes (B, T, Tregs, macrophages, and Lin⁻ cells), as well as increased cytokines (IL-4, IL-6, IL-12, and IL-10), suggesting a systemic stimulatory effect in co-colonized animals. The importance of the early-life bacterial and fungal microbiome was further supported by studies in mice treated with antimicrobials, with antibiotic- and antifungal-treated mice displaying similar trends to the Y-only and B-only groups, respectively (Fig. 5a, b, Supplementary Fig. 6a, b). Of note, the substantial immune changes were not due to systemic fungal infection in any of the groups, as corroborated by negative fungal cultures in kidney, a feature of fungemia. This is in line with a recent study in human T memory cells specific to *C. albicans*, which displayed cross-reactivity toward *Aspergillus fumigatus* in the airway[39], suggesting that gut fungi colonization is important for immune education and immunological memory toward fungal exposures at other mucosal sites.

Similarities in cytokine response between GF and Y groups were also observed following 5 days of treatment with DSS, in which absence of microbes or exclusive fungal colonization resulted in reduced inflammation. Our results support previous work showing that bacterial colonization is essential for the development of DSS colitis[40], and expand current knowledge by demonstrating that exclusive colonization with this fungal consortium had little to no effect on colitis induction. In line with previous studies, our model showed higher expression of IL-17A

in intestinal tissues in bacteria-colonized groups following DSS treatment (Fig. 2e). IL-17A is a potent enhancer of DSS-induced colitis phenotype, with elevated IL-17A secreted by Th17 cells promoting secretion of IL-12 and IL-23, and triggering a conversion of the response from Th17 to Th1[41,42]. Our model showed that bacteria colonization induced elevated detection of Th1 pro-inflammatory cytokines INF-γ, IL-6, and TNF-α, which have been previously suggested to create an inflammatory loop and worsen colitis in mice[43]. In addition, our work convincingly showed that IL-22 secretion only occurred in exclusive colonization with bacteria, and that presence of fungi prevented this (Fig. 2e), which may help explain the changes in inflammatory response attributed to fungal colonization. IL-22 plays dual roles in intestinal inflammation. For instance, T-cell production of IL-17 and IFN-γ is associated with a deleterious function of IL-22[44], yet colonic delivery of IL-22 gene via a secretory vector increased goblet cell expression and ameliorated DSS-induced colitis[45]. In this context, the prevention of IL-22 secretion in mouse groups colonized with fungi (Y and BY) may be a contributing factor to the elevated inflammatory response in co-colonized mice (BY). Further research testing the modulatory capacity of IL-22 in gnotobiotic conditions will be necessary to confirm this.

Several studies have reported that fungal overgrowth exacerbates DSS colitis in SPF mice[15,46,47]. However, in light of our observations that fungal signals alone are unable to induce the level of colonic inflammation commonly reported in DSS-treated animals (Fig. 2), it is more likely that interactions between bacteria and fungi lead to exacerbated colitis. A recent study in mice provides convincing evidence that CARD9 signaling via Dectin-2 receptor activation in *Malassezia*-treated animals worsens colitis[20]. In their work, Limon et al.[20] demonstrated that CARD9 activation induced a pro-inflammatory cascade toward innate immune cell activation and exacerbation of intestinal inflammation. Further research is needed to confirm if this activation is also dependent on the concomitant presence of bacterial signals, as our results suggest, or if it is driven by specific fungal species, as our fungal consortium did not include *Malassezia* sp.

Our results also revealed that colonization with bacteria (B and BY) but not exclusive fungal colonization reduced OVA-induced airway inflammation in mice, and that fungi promote infiltration of macrophages/monocytes in the bronchoalveolar space (Fig. 6), coinciding with our findings on fungi-driven early-life changes in splenic cell populations (Fig. 5). A possible mechanism may involve prostaglandin E₂ (PGE2), a strong regulator of macrophage polarization produced and secreted by *Candida* species. This is supported by previous work from Kim et al.[22], showing that antibiotic-induced fungal overgrowth in the gut led to increased M2-macrophage infiltration and PGE2 plasma concentrations. These findings show that bacterial signals early in life are critical to prevent increased susceptibility to allergic airway inflammation, and that fungal colonizers can act as modulators of the underlying immune profile of airway inflammatory responses. Our work provides a clear portrayal of how the initial composition of the gut microbiome, as well as perturbations to it, distinctly dictate the strength and cell types involved in the immune response toward a mucosal antigen.

Our study did not interrogate the fungal-driven mechanisms causing the above-mentioned immune changes. We did not detect any metabolites that differed between the BY from the B-only groups. Therefore, we speculate that these effects are not driven by the uptake of fungal-derived metabolites by immune cells. An alternative, more plausible mechanism may involve direct interactions between fungal-associated molecular patterns with immune cells. Further mechanistic work evaluating the effect of specific structural and/or secreted fungal molecular patterns

has great potential to reveal microbial patterns that may act as specific triggers of these reported immune responses.

Overall, our unique experimental approach enabled the demonstration that fungal colonization significantly impacts early-life host immune development and susceptibility to inflammatory-disease in the distal gut and the lungs. This work causally implicates fungi as microorganisms that can skew the immature immune system and modulate susceptibility to immune-mediated disorders later in life. Further efforts are required to fully understand the specific implications of fungal alterations and, more importantly, how to prevent their manifestations. This ecology-driven framework will help inform the design of future microbiome strategies to remediate early-life microbial alterations by identifying fungal taxa with beneficial effects on both the developing microbiome and host immunity.

## Methods

**Colonization of gnotobiotic mice.** Ten- to fifteen-week-old GF C57Bl/6J mice were obtained from the gnotobiotic mouse facility of the International Microbiome Center (IMC) at the University of Calgary. Female mice were orally gavaged twice, 3 days apart, with 100 μl of a consortium of microorganisms, or kept germ-free. Colonization consisted of consortia of (i) 12 mouse-derived bacteria, "B"[27] (*Acutalibacter muris*, *A. muciniphila*, *Bacteroides caecimuris*, *B. longum* subsp. *animalis*, *B. coccoides*, *C. clostridioforme*, *C. innocuum*, *E. faecalis*, *F. plautii*, *L. reuteri*, *M. intestinale*, and *T. muris*), (ii) five yeast species previously linked to atopy and asthma risk in infants, "Y"[12,13] (*C. albicans*, *C. glabrata*, *C. parapsilosis*, *I. orientalis*, and *R. mucilaginosa*), or (iii) a combination of all 17 bacterial and yeast species, "B + Y" (source of microbes used in Supplementary Table 1). Inocula were prepared under anaerobic conditions by mixing 100 μl of 2-day-old microbial cultures of each species. Bacteria were grown in fastidious anaerobe broth (LabM, Heywood, Lancashire, UK) and yeasts were grown in yeast-mold broth (YM; BD, Sparks, MD, USA). After the second gavage, mice were paired for mating on a 2:1 female:male ratio per cage. To ensure microbial colonization with the desired consortia in F1 mice, the corresponding inocula were further spread on the dams abdominal and nipple regions on days 3 and 5 after birth, as previously described[48]. Two groups of mice colonized with both bacteria and yeasts were also treated with the antibiotic Augmentin (0.2 mg/ml; Sigma, Oakville, ON, Canada) or antifungal Fluconazole (0.5 mg/ml; Sigma) in sterile drinking water from day 7 to 14 after birth. Treatment solutions were prepared by dissolving the antimicrobials in distilled water, followed by filter sterilization. Mice were kept at maximum five animals per isolator cage and housed inside gnotobiotic flexible-film isolators in the IMC, under a 12-h light/12-h dark cycle, 40% relative humidity, 22–25 °C, and ad libitum access to sterile food and water. All experiments were conducted under protocols approved by the University of Calgary Animal Care Committee, following the guidelines of the Canadian Council on Animal Care.

**Fungal culture in selective medium.** Fungal colonization was assessed by culture in selective medium. Mice fecal pellets (~20 mg) were dissolved in 500 μl of 0.9% sterile saline, serially diluted tenfold, and 100 μl of each dilution was spread over YM agar plates supplemented with gentamycin (10 μg/ml; Sigma) and chloramphenicol (200 μg/ml; Sigma). Plates were incubated at 25 °C and checked for yeast-like colonies for up to 10 days. Viable fungal concentration was determined by plate count method of colonies.

**Fungal and bacterial quantification (qPCR assays).** Fungal and bacterial DNA concentrations in feces were determined by quantitative amplification of the fungal 18S and bacterial 16S rRNA genes, as previously described[12,49,50]. Gut microbial DNA was obtained from fecal samples using the DNeasy PowerSoil Pro kit (Qiagen, Hilden, Germany) according to the manufacturer's instructions, eluted in nuclease-free water, and stored at −80 °C. All qPCR reactions were carried using iQ SYBR Green Supermix (BioRad Laboratories, Hercules, CA, USA). For the 16S qPCR protocol, 10-μl reactions consisted of 5 μl master mix, 0.5 μl of each primer (u16Sfr and u16Srv; see Supplementary Table 1) at 10 μM, 2 μl nuclease-free water, and 2 μl of 1 ng/μl dilution of extracted template DNA. The 16S qPCR thermocycler program consisted of an initial 5 min step at 94 °C, followed by 40 cycles of 94 °C for 15 s, 60 °C for 30 s, and 72 °C for 30 s, and a final melt curve consisting of a single cycle of 95 °C for 15 s, 60 °C for 1 min, 95 °C for 15 s, and 60 °C for 15 s. For the fungal 18S qPCR protocol, 20-μl reactions consisted of 10 μl master mix, 2.5 μl of each primer (FR1 and FF390; see Supplementary Table 1) at 10 μM, 3 μl nuclease-free water, and 2 μl of 1 ng/μl dilution of template DNA. The fungal 18S qPCR thermocycler program consisted of an initial 10 min step at 95 °C, followed by 40 cycles of 95 °C for 15 s, 50 °C for 30 s, and 70 °C for 60 s, and a final melt curve consisting of a single cycle of 95 °C for 15 s, 60 °C for 1 min, 95 °C for 15 s, and 60 °C for 15 s. All samples were run in duplicate and concentration calculated based on standard curves generated using bacterial genomic DNA (extracted with DNeasy PowerSoil Pro kit; Qiagen) or fungal genomic DNA (extracted with

YeaStar Genomic DNA Kit; Zymo Research, Irvine, CA, USA). qPCR runs were performed on the StepOne Plus Real-Time PCR System (Applied Biosystems, Foster City, CA, USA) in the Snyder Resource Laboratories, University of Calgary.

**Fecal water content**. Fecal pellets were collected and immediately weighed (5–10 mice/treatment group). Pellets were dried overnight at 50 °C, reweighed to determine wet:dry weight ratios.

**Gastrointestinal barrier assessment and small intestinal transit**. Intestinal permeability was measured using previously described methods with some modifications[51]. Briefly, mice (4–5/treatment group) were gavaged orally with 100 µl of 50 mg/ml fluorescein-5(6)-sulfonic acid, trisodium salt (FSA, 478 daltons; Setareh Biotech, LLC, Eugene, OR, USA). Small intestinal transit was assessed as previously described in detail[52]. Briefly, 3 h 45 min following FSA gavage mice were gavaged with 200 µl Evans Blue (5% suspended in 5% Gum Arabic; Sigma). Four hours after FSA gavage and 15 min after Evans Blue gavage animals were anesthetized with isofluorane and blood was drawn by cardiac puncture for serum FSA content. Blood was allowed to clot at room temperature, spun at $2000 \times g$ for 10 min, and the supernatant (serum) read on a spectrophotometer at 485/535 nm. Serum FSA concentration (µg/ml) was determined from a standard curve. To assess small intestinal transit, animals were euthanized by cervical dislocation and the entire gastrointestinal tract from the stomach to the anus was immediately removed. The distance traveled by the colored marker was measured and expressed as a percentage of the total small intestinal length from the pylorus to the cecum. As described below, cecal weight, small intestinal, and colon length were determined, and tissue from the colon was used to conduct ex vivo measurement of ion secretion and transepithelial resistance.

**Colonic propulsion, cecal weight, and small intestinal and colon length**. Distal colonic propulsion was measured as previously described[53], with slight modifications. Animals (4–5/treatment group) were allowed to acclimate for 30 min after being transferred to the lab. Mice were lightly anesthetized with isofluorane before a plastic bead (2.5 mm diameter) was inserted 3 cm into the distal colon using a silicone pusher. The time to expulsion of the bead was determined in seconds. Following bead expulsion mice were anesthetized with isofluorane, killed by cervical dislocation, and the entire gastrointestinal tract from the stomach to the anus was immediately removed for measurement of cecal weight and small intestinal and colon length. Tissue from the ileum was used to conduct ex vivo measurement of ion secretion and transepithelial resistance as described below. For each mouse killed (8–10/treatment group) the full cecum was weighed and then the contents removed, and the empty cecum reweighed. Both the length of the ileum, from the pylorus to the cecum, and the colon length were measured.

**Measurement of ion secretion and transepithelial resistance**. Full-thickness segments of terminal ileum, proximal colon, and distal colon were opened along the mesenteric border, cleaned of luminal contents, and mounted in Ussing Chambers (Physiologic Instruments, San Diego, CA, USA) with an exposed area of 0.3 cm$^2$. The tissues were bathed at 37 °C, in oxygenated (95% $O_2$, 5% $CO_2$) Krebs solution (pH 7.4) with 10 mM of glucose and mannitol in the serosal and mucosal compartments, respectively. Tissues were held under voltage-clamp conditions (0 V), and allowed to equilibrate for 15–20 min after which basal short-circuit current (Isc, µA/cm$^2$) and transepithelial potential were measured in order to calculate basal transepithelial resistance according to Ohm's law (TER, Ω/cm$^2$). Changes in net electrogenic ion flux were evaluated in the ileum by measuring changes in Isc in response to neuronal depolarization with 10 µM veratridine (Calbiochem, San Diego, CA, USA) or in response to muscarinic receptor stimulation with 100 µM carbachol (Sigma). The difference between basal Isc and peak Isc recorded after veratridine or carbachol addition was measured (ΔIsc, µA/cm$^2$). A positive ΔIsc indicated a luminally directed negative net charge transfer (anion secretion). Measurements were conducted and averaged in two adjacent intestinal segments from the same mouse for every gut region (4–5 mice/treatment group).

**DSS-colitis model**. DSS-induced colitis model was performed as previously described[54], with small modifications. Briefly, F1 mice at 7 weeks of age (7–11/treatment group) were treated for 5 consecutive days with 1.5% DSS (Alfa Aesar, Haverhill, MA, USA) in sterile drinking water. DSS solution was prepared by dissolving the colitogenic agent in distilled water, followed by filter sterilization. DSS consumption was monitored daily by determination of water volume consumed per cage. Health checks and body weight measurements were performed daily to follow disease progression. Feces were obtained for assaying fecal markers of inflammation. Occult blood was determined in Hemoccult Fecal Occult Blood Slide Test System (Beckman Coulter, Brea, CA, USA) and lipocalin-2 (Lcn-2) protein was assayed in Mouse Lipocalin-2/NGAL DuoSet ELISA (R&D Systems, Minneapolis, MN, USA). At the end of DSS treatment, mice were anesthetized with isofluorane and sacrificed by cervical dislocation. Sections of large intestines were obtained for histology and flash-frozen for cytokine measurements in U-Plex T Cell Combo Assay Kit in MESO QuickPlex SQ 120 (Meso Scale Discovery, Rockville, MD, USA). Sections for histology were fixed in Neutral Buffered Formalin 10% (EMD Chemicals, Gibbstown, NJ, USA) and embedded in paraffin. Histological

cuts were stained with hematoxylin and eosin for inflammation scoring, as described below. Flash-frozen tissues were weighed and lysed in 600 µl of lysis buffer, constituted of 150 mM sodium chloride (EMD Chemicals), 20 mM Tris-Hydrochloride pH 7.5 (EMD Chemicals), 1 mM EGTA (ethylene glycol-bis(β-aminoethyl ether)-N,N,N′,N′-tetraacetic acid; Sigma), 1% Triton X-100 Surfactant (EMD Chemicals), and 1x cOmplete Protease Inhibitor Cocktail (Roche, Basel, Switzerland). Protein concentration in tissue lysates was determined using the Coomassie (Bradford) Protein Assay Kit (Thermo-Fisher, Waltham, MA, USA).

**Colonic histologic scoring**. Inflammation was blindly assessed following modified protocol from Chassaing et al.[54] Each section was assigned scores based on the following parameters: (i) inflammatory infiltration (0 = none, 1 = inflammatory cells above muscularis serosa only, 2 = inflammatory cells in submucosa, 3 = inflammatory cells in muscularis serosa to muscularis mucosa, or 4 = extensive inflammation in mucosa and epithelial layer), (ii) crypt damage (0 = none, 1 = <30%, 2 = <60%, 3 = only epithelial surface intact, or 4 = entire crypt and epithelia lost), (iii) goblet cell depletion (0 = none or 1 = present), and (iv) crypt abscess (0 = none or 1 = present). In addition, parameters "i", "ii", and "iii" were further multiplied by a degree factor of 1 = focal, 2 = patchy, or 3 = diffused. Total score ranged from 0 to 28 per mouse.

**DNA library preparation and sequencing**. Fecal microbial DNA extracted with DNeasy PowerSoil Pro Kit (Qiagen) was used to amplify the V4 region of the bacterial 16S rRNA gene and the ITS2 region of the fungal ITS genetic marker. This generated ready-to-pool dual-indexed amplicon libraries as described previously[55,56]. 16S amplicon libraries were prepared in-house using TopTaq Master Mix kit (Qiagen). Amplicons were cleaned up using QIAquick PCR purification kit (Qiagen), quantified with PicoGreen (Invitrogen, Carlsbad, CA, USA), and diluted to 20 ng/µl for sequencing. ITS2 amplicon libraries were prepared at Microbiome Insights (University of British Columbia, UBC, Vancouver). In-house extracted DNA samples were sent to the facility and amplified using Phusion Hot Start II DNA Polymerase (Thermo-Fisher). PCR reactions were cleaned up, normalized using the high-throughput SequalPrep Normalization Plate Kit (Applied Biosystems), and quantified accurately with the KAPA qPCR Library Quantification kit (Roche). Controls without template DNA, and mock communities with known amounts of selected bacteria and fungi, were included in the PCR and downstream sequencing steps to control for microbial contamination and verify bioinformatics analysis pipeline. The pooled and indexed libraries were denatured, diluted, and sequenced in paired-end modus on an Illumina MiSeq (Illumina Inc., San Diego, USA). 16S and ITS2 sequencing were performed at Microbiome Insights.

**Microbiome analyses**. Sequences were checked for quality, trimmed, merged, and checked for chimeras using the DADA2 v1.10.1[57] pipelines for 16S or ITS2 and phyloseq v.1.26.1[58] as packages for R (R Development Core Team; http://www.R-project.org) in RStudio v.1.1463. We built bacterial and fungal community matrices from the resulting unique ASV based on the UNITE v.8.0 (fungi)[59] and SILVA v.132 (bacteria)[60] databases. Complete workflow tutorials can be found at https://benjjneb.github.io/dada2/tutorial.html and benjjneb.github.io/dada2/ITS_workflow.html. For community-level analyses (alpha- and beta-diversity and relative abundance), the most abundant 22 bacterial and 16 fungal ASVs, which explained >98% of all assigned reads, were merged by species name (refer to Source data file).

**Species-specific qPCR**. C. albicans and R. mucilaginosa were quantified using species-specific qPCR in DNA samples extracted from feces of 4-week-old mice, as previously described[61,62], with small modifications. For the C. albicans-specific qPCR, 20-µl reactions consisted of 10 µl master mix, 1 µl of each primer (CALB1 and CALB2; see Supplementary Table 1) at 10 µM, 6 µl nuclease-free water, and 2 µl of 1 ng/µl dilution of extracted template DNA. The C. albicans qPCR thermocycler program consisted of an initial 10 min step at 95 °C, followed by 35 cycles of 95 °C for 30 s and 60 °C for 1 min, and a final melt curve consisting of a single cycle of 95 °C for 15 s, 65 °C for 1 min, and 95 °C for 15 s. For the R. mucilaginosa-specific qPCR protocol, 15-µl reactions consisted of 7.5 µl master mix, 0.5 µl of each primer (RM-5fw and RM-3bw; see Supplementary Table 1) at 10 µM, 4.5 µl nuclease-free water, and 2 µl of 1 ng/µl dilution of DNA template. The R. mucilaginosa qPCR thermocycler program consisted of an initial 2 min step at 50 °C, followed by another 2 min step at 95 °C, and 40 cycles of 95 °C for 15 s and 60 °C for 45 s, and a final melt curve consisting of a single cycle of 95 °C for 15 s, 63 °C for 1 min, and 95 °C for 15 s. Samples were run in duplicate and concentration calculated based on standard curves generated using C. albicans or R. mucilaginosa genomic DNA (extracted with YeaStar Genomic DNA Kit; Zymo Research). qPCR runs were performed on the StepOne Plus Real-Time PCR System (Applied Biosystems) in the Snyder Resource Laboratories, University of Calgary.

**Metabolomics**. Metabolites were extracted from gnotobiotic mice fecal pellets and small intestinal contents by 50% methanol solution. Samples were subjected to untargeted metabolomics analysis using LC-MS. Metabolomics data were acquired at the Calgary Metabolomics Research Facility, which is supported by the IMC. General

metabolomics runs were performed on a Q Exactive HF Hybrid Quadrupole-Orbitrap Mass Spectrometer (Thermo-Fisher) coupled to a Vanquish UHPLC System (Thermo-Fisher). Chromatographic separation was achieved on a Syncronis HILIC UHPLC column (2.1 mm × 100 mm × 1.7 μm; Thermo-Fisher) using a binary solvent system (A + B) at a flow rate of 600 μL/min. Solvent A, 20 mM ammonium formate pH 3.0 in mass spectrometry grade $H_2O$; Solvent B, mass spectrometry grade acetonitrile (Thermo-Fisher) with 0.1% formic acid (%v/v). A sample injection volume of 2 μL was used. The mass spectrometer was run in negative full-scan mode at a resolution of 240,000 scanning from 50 to 750 $m/z$.

**Early-life immune assessment**. Four-week-old F1 mice (5–8/treatment group) were selected for early-life immune assessment. Mice were anaesthetized with isoflurane and blood was obtained by cardiac puncture after confirmed absence of reflex. Following cardiac puncture, mice were sacrificed by cervical dislocation and spleen, jejunum, and colon sections were immediately removed for flow cytometry or cytokine measurements as described below. Blood was allowed to clot on ice, spun at $3000 \times g$ for 10 min, and supernatant (serum) was used to determine immunoglobulin levels. IgE titer was determined in IgE BD OptEIA ELISA (BD Biosciences, San Diego, CA, USA). IgA, IgG1, IgG2a, IgG2b, IgG3, and IgM titers were determined in Mouse Isotyping Panel 1 Assay Kit (Meso Scale Discovery). Jejunum and colon fragments were flash-frozen for cytokine measurements in U-Plex T Cell Combo Assay Kit (Meso Scale Discovery).

**Immunolabeling and flow cytometry**. Spleens were surgically removed and processed for flow cytometry. Spleens were kept in ice-cold RPMI 1640 + Gluta-Max-I medium (Thermo-Fisher) and were macerated in GentleMACS C tubes (Miltenyi Biotec, Bergisch Gladbach, Germany). Cellular macerate was filtered through a 100 μm cell strainer and red blood cells were lysed in 5 ml 1x RBC Lysis Buffer (BioLegend, San Diego, CA, USA). Splenocytes were resuspended in ice-cold cell growth media constituted of RPMI 1640 + GlutaMax-I medium (Thermo-Fisher) supplemented with 10% heat-inactivated Fetal Bovine Serum (FBS; Thermo-Fisher), 2000 U Penicillin-Streptomycin (Thermo-Fisher), and 50 μM 2-Mercaptoethanol (Thermo-Fisher). Isolated splenocytes were stained with Fixable Viability Stain (FVS575V; BD Biosciences) and array of intra and extracellular markers for immune cell characterization. Cells were stained for surface markers, fixed and permeabilized with BD Transcription Factor Buffer Set (BD Biosciences), and stained for intracellular markers. The following antibodies were used: PE-Cy7 anti-mouse CD19 (clone 1D3; BD Biosciences), PerCP-Cy5.5 anti-CD11b (M1/70; BD Biosciences), APC-H7 anti-mouse CD8a (53-6.7; BD Biosciences), Alexa Fluor 700 anti-mouse CD3 (17A2; BD Biosciences), V500 anti-mouse CD4 (RM4-5; BD Biosciences), Brilliant Violet 421 anti-mouse CD25 (PC61; BD Biosciences), Brilliant Violet 421 anti-mouse CD11c (HL3; BD Biosciences), PE anti-Gata3 (L50-823; BD Biosciences), PE anti-mouse IL-17A (TC11-18H10; BD Biosciences), PE anti-mouse IL-12/23 (C15.6; BD Biosciences), PE anti-mouse F4/80 (T45-2342; BD Biosciences), Alexa Fluor 488 anti-mouse IL-4 (11B11; BD Biosciences), Alexa Fluor 488 anti-mouse FoxP3 (MF23; BD Biosciences), Alexa Fluor 488 anti-mouse INF-γ (XmG1.2; BD Biosciences), Alexa Fluor 488 anti-mouse I-A/I-E (M5/114.15.2; BD Biosciences), eFluor 660 anti-mouse IL-13 (eBio13A; eBioscience, San Diego, CA), APC anti-mouse IL-10 (JES5-16E3; BD Biosciences), APC anti-mouse IL-6 (MP5-20F3; BD Biosciences), and APC anti-mouse CD103 (M290; BD Biosciences). All antibodies for surface and intracellular markers were diluted 200 times, with the exception of CD8a that was diluted 100 times. Samples were run in BD FACSCanto II (BD Biosciences), in the Nicole Perkins Microbial Communities Core Lab, Snyder Institute for Chronic Diseases, University of Calgary.

**OVA-induced airway inflammation model**. An experimental murine OVA model was followed as previously described[48] with modifications. F1 mice (5–10/treatment group) were systemically sensitized intraperitoneally (IP) with sterile 200 μg of grade V OVA (Sigma) and 1.3 mg of aluminum hydroxide (Thermo-Fisher) at weeks 7 and 9 post-birth. Following systemic sensitization, airway inflammation was induced by intranasal challenges at week 10 with 50 μg of Low Endo OVA (Whortington, Lakewood, NJ, USA) for 3 consecutive days, followed by 2 days of 100 μg of grade V OVA (Sigma) in sterile phosphate-buffered saline (PBS). Intranasal challenges were done in mice under anesthesia with isoflurane. Following challenge scheme, mice were anesthetized with ketamine (200 mg/kg; Vetoquinol, Lavaltrie, QC, Canada) and xylazine (10 mg/kg; Bayer Inc., Mississauga, ON, Canada), and BAL were performed by 3 × 1 ml washes with PBS + 10% FBS for total and differential cell counts. Total BAL counts were performed in hemocytometer. Differentials (eosinophils, neutrophils, macrophages, and lymphocytes) were performed from 200 cells in hematoxylin and eosin stained CytoSpins (Shandon Cytospin II, Thermo-Shandon, Runcorn, Cheshire, UK) based on standard morphological criteria.

**Data visualization and statistical analysis**. We measured gut bacterial alpha-diversity using the Chao1 (richness) and Shannon indices. Non-parametric Mann–Whitney tests were performed to test for significant differences in alpha-diversity between treatments and time points. To account for potential hetero-skedasticity in community beta-diversity dispersion and avoid the loss of

information through rarefaction[63], a variance stabilizing transformation was performed prior to any statistical tests[63,64]. Changes in gut bacterial community structure (beta-diversity) were assessed statistically using PERMANOVA[30] and visualized using principal coordinate analysis (PCoA) based on Bray–Curtis dissimilarities. To explore further the changes in taxonomic community structure, significant changes in relative abundance of all microbial species were tested using non-parametric Kruskal–Wallis tests followed by post hoc Dunn tests with Benjamin–Holmes with FDR correction. Boxplots were made to identify outliers and normality was assessed using the Shapiro–Wilk test. If the data was normally distributed, parametric one-way analysis of variance (ANOVA) with Tukey's post hoc tests was used. Correlations between the merged 16S and ITS2 ASVs and the detected immunological features in early life were assessed using the BiCOR method with FDR correction. We also used SGCCA to understand the correlation structure between different measurements (16S, ITS2, and immune features) while having the possibility to use dimension reduction especially for the highly correlated metabolomics data. The R package mixomics[65] was used to perform SGCCA analysis. ITS2 and 16S data were first changed to ranks[57] and all measurements were range normalized. For longitudinal analyses, statistical tests on univariate response variables were performed using a linear mixed-model for repeated measures, followed by an ANOVA with Tukey's post hoc when appropriate. PCA, ANOVA, T-tests, and pathway analysis of fecal metabolites originating from different mice groups was performed by using Metaboanalyst v.4.0[66]. In all tests, significance was set at $P < 0.05$. All data points represent biological replicates. Bar graphs display mean with standard error of the mean (S.E.M.). Boxplots of alpha-diversity, relative abundance, and normalized metabolite concentration display median with hinges and whiskers, the outlying points beyond end of whiskers are plotted individually. Lower and upper hinges show data 25th and 75th percentile, respectively. Whiskers extend from hinge to the lowest/largest value bellow 1.5 times the inter-quartile range. Graphs were made using either RStudio (R Development Core Team, v. 1.1.463), Metaboanalyst version 4.0 (Xia Lab, McGill University, Montreal, QC, Canada), or Prism version 8.1.2 (GraphPad Software, La Jolla, CA, USA). Figures 1, 2, and 6 were created using diagrams from BioRender.com (https://biorender.com/). Post hoc comparisons were applied to determine significant differences between all treatment groups. Due to the large number of comparisons we used a letter system to denote statistical significance. The absence of significant differences between groups is indicated by these groups sharing the same letter above their bars.

**Reporting summary**. Further information on research design is available in the Nature Research Reporting Summary linked to this article.

## Data availability

The source data underlying all presented Figures and Supplementary Figures and Tables are provided as a Source Data file. 16S and ITS2 sequence reads and supporting metadata were deposited to European Nucleotide Archive (ENA), https://www.ebi.ac.uk/ena/browser/view/PRJEB35163 (study accession numbers PRJEB35163/ERP118175). Metabolomics mass spectral raw data were deposited to MetaboLights, https://www.ebi.ac.uk/metabolights/MTBLS1679 (study identifier MTBLS1679). All microbial strains used are kept as stocks in the Arrieta lab and can be shared when requested.

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

## Acknowledgements

The authors would like to acknowledge the following collaborators for their support: Markus Geuking, Simon Hirota, and Jens Walter for conceptual feedback and/or help with study design; Dylan Pillai and Alberta Public Laboratories for the fungal isolates; Laurie Wallace, Cristiane Baggio, and Kristoff Nieves for technical support; Karen Poon,

Matheus Heitor, and Carolyn Thomson for their help with flow cytometry approaches; Kristy Brown, Ian Lewis, and Ryan Groves for metabolomics support. This work was supported by funds from the Cumming School of Medicine, the Alberta Children Hospital Research Institute, the Snyder Institute of Chronic Diseases, the Canadian Institutes for Health Research, the Sick Kids Foundation, the W. Garfield Weston Foundation, The Koopmans Research Fund, and the Canadian Lung Association. K.A.S. holds the Crohn's and Colitis Canada Chair in IBD Research. E.v.T.B. is funded by the Eyes High Doctoral Recruitment Scholarship. V.K.P. is financed by the Research Council of Norway FRIPRO Mobility Research Grant, which is co-funded by the European Union's Seventh Framework Program for research, technological development, and demonstration under Marie Curie grant. N.G.J. is funded by the Parker B Francis Fellowship, Alberta Innovates, and NSERC BRAIN CREATE. J.-B.C. is funded by the Human Frontier Science Program. F.A.V. is funded by the National Council for Scientific and Technological Development (CNPq/Brazil). The IMC is supported by the Cumming School of Medicine, University of Calgary, Western Economic Diversification (WED), and Alberta Economic Development and Trade (AEDT), Canada.

## Author contributions

E.v.T.B., V.K.P., M.W.G., I.L.L., K.D.M., W.K.M., K.A.S., and M.-C.A. contributed to study design; E.v.T.B., V.K.P., M.W.G., I.L.L., and M.-C.A. carried out the mouse experiments; E.v.T.B., V.K.P., M.W.G., I.L.L., N.G.J., J.-B.C., F.A.V., C.M.K., J.S., and M.-C.A. prepared all the samples; E.v.T.B., M.W.G., and M.-C.A. optimized the qPCR strategy; E.v.T.B., M.W.G., and M.-C.A. performed the qPCR analysis; J.-B.C., F.A.V., and C.M.K. performed and analyzed gut physiology experiments; E.v.T.B., M.W.G., and M.-C.A. analyzed DSS-colitis data; E.v.T.B., I.L.L., J.S., and M.-C.A. prepared libraries and optimized sequencing strategy; E.v.T.B., I.L.L., and M.-C.A. analyzed the 16S and ITS2 data; V.K.P. and M.-C.A. analyzed the metabolomics data; E.v.T.B. and M.-C.A. analyzed the early-life immunity data; E.v.T.B., M.W.G., N.G.J., R.J.A.W., and M.M.K. analyzed airway inflammation data; E.v.T.B., V.K.P., M.W.G., I.L.L., and M.-C.A. curated all the metadata; H.R. performed SGCCA for multi-omics correlation; E.v.T.B. and M.-C.A. wrote the paper; E.v.T.B., M.W.G., V.K.P., N.G.J., K.D.M., C.M.K, F.A.V., J.-B.C., K.A.S., and M.-C.A. edited the paper. All authors contributed extensively to the work presented in this paper.

## Competing interests

The authors declare no competing interests.
