## [Peer Review File · Nature Communications]

Reviewers' comments:

Reviewer #1 (Remarks to the Author):

This is an important and well-written experimental study on the role of fungi in mouse intestines. It is clear that a lot of work has gone into this study.

Line 1. I'd prefer a more informative title. Right now, the title says nothing about the results.

63-64. The authors are critical that other researchers routinely ignore fungi in microbiome studies, and I agree. Yet the authors fall into the same trap that everyone falls into – highlighting human-associated fungi as solely pathogenic. The authors thus give an unbalanced account of the role of fungi in human health and disease. See, e.g., <https://journals.plos.org/plospathogens/article?id=10.1371/journal.ppat.1002808>, <https://www.ncbi.nlm.nih.gov/pubmed/28102762>, and <https://www.ncbi.nlm.nih.gov/pmc/articles/PMC5840654/> for various non-pathogenic, probiotic, and evolutionary examples.

78. Remarkably for a study on fungi, none of the authors seem to be a mycologist (even though Canada is full of mycologists). This shows over and over again in the manuscript. On this line, "*Pichia kudriavzevii*" is an obsolete synonym of "*Issatchenkia orientalis*". See Index Fungorum (<http://www.indexfungorum.org/Names/Names.asp>).

102. How can the authors know that all of the fungi in their study are "commensal"? This strikes me as very unlikely. It is also hard for me to assess, since the authors withhold most of their primary data (see below). If the authors want to classify the fungi recovered into functional guilds, then FUNGuild (<https://www.sciencedirect.com/science/article/abs/pii/S1754504815000847>) is readily available. I find it hard to come up with a reason why the authors chose not to use FUNGuild to build a functional profile of the fungi recovered.

180. The ITS region does not formally qualify as a "gene", so on this line, "gene sequencing" is not formally correct. I propose "marker sequencing".

188. The abbreviation "PERMANOVA" is defined twice in the manuscript. I'd say once is enough?

226. The authors withhold the list of fungal [and, for that matter, bacterial] species recovered (presumably for a second publication). Not releasing the full set of results is not acceptable if you ask me. Please provide:

- * a summary pie chart showing the taxonomic affiliations of all fungi [and bacteria] recovered

- * a FUNGuild chart of the functional guilds of the fungi recovered

- * a supplementary item with all ASVs and their taxonomic affiliation, including the DOI of each ASV as assessed through comparison to the UNITE database (<https://unite.ut.ee/repository.php>). Most of the ASVs of the authors will presumably lack full species names, so providing a DOI is the only way to make the taxonomic results comparable across studies and time. Failure to do so would compromise the scientific reproducibility of the results.

230. “taxa” should be “taxon”. Singular noun. Or are the authors somehow implying that this name is non-monophyletic? If so, please back it with at least one solid recent reference.

237. What is the reader supposed to get from “(bicolor method, FDR corrected)” at this stage in the manuscript?

238-242. Please write the species names out in full. Most readers will have no clue what species are intended otherwise.

258. “Heat-map” > “The heap-map” ?

271. Most of the time, the authors use the Oxford comma when it comes to enumerations (e.g., 327 and 329). But on this line and, e.g., 272, they don’t. Please resolve this in a consistent way throughout the manuscript.

279. “yielded specific grouping of the mice” sounds awkward. Please revise for clarity.

369-373. This part is very interesting. Should it be given a more prominent role in the manuscript (including the abstract)?

385. Can this really be true? After all, the fact that there are fungi that are specialized in colonizing mammalian intestines has been known for 100+ years?
(<https://en.wikipedia.org/wiki/Neocallimastigomycota> ;
<https://mycokeys.pensoft.net/article/28337/>)

395. Again the authors focus solely on the negative aspects of fungi. There are many examples of symbioses between fungi and bacteria – such as lichens and, e.g.,
<https://www.ncbi.nlm.nih.gov/pmc/articles/PMC3232736/> - but the authors conveniently ignore those. Also, please provide a (modern) reference to back the claim that most fungal-bacterial interactions in the soil take the form of “competition and antagonism”. The authors will probably struggle to come up with such a reference.

435. Please clarify the “determining...” statement. I have a hard time making sense out of it.

462. The “sp.” should not be given in italics.

526. “For 16S” > “For the 16S” ?

531. “For fungal” > “For the fungal” ?

629. Of all 16S regions the authors could have chosen, the V4 region is the least informative one from a taxonomic point of view (see Fig. 1 in
<https://www.sciencedirect.com/science/article/pii/S0167701210002745?via=ihub>). Why was this poorly performing region chosen to begin with?

629. There are no V regions in the fungal ITS region. So what the authors mean by “V3-V4” here is beyond me. Is this a simple copy-and-paste mistake from the 16S part?

630. Like I said before, the ITS region does not formally qualify as a “gene”. Please use “marker” or “genetic marker” instead.

647. The metabarcoding part leaves a lot to be desired in terms of reproducible science. Please provide considerably more information here. What pipeline was used to process the raw data? What ASV parameters were used? Why wasn't LULU (<https://www.nature.com/articles/s41467-017-01312-x>) used? What taxonomic reference databases were used? Which software tools were used, and what were their version numbers? Where can the reader access the raw sequence data and the OTU tables? I presume this is the first time the authors report on metabarcoding results – please see, e.g., <https://academic.oup.com/gigascience/article/4/1/s13742-015-0074-5/2707574> and <https://nph.onlinelibrary.wiley.com/doi/10.1111/j.1469-8137.2011.03755.x> on how to do it.

734. “R package” > “The R package”. Also, please provide the version number of R and all other software tools and packages used in the study. Failure to do so would compromise scientific reproducibility (which is already quite low in this study, owing to the authors' withholding the raw data, the OTU (ASV) tables, the full taxonomic results, and so on).

748. “fungi” > “fungal”

778. What, exactly, was deposited here? Please deposit all relevant files, including the raw sequence data, in ENA or SRA. And please follow standards by also providing the deposition accession number.

818 and elsewhere. The list of references is heterogeneously specified. In this particular reference, article titles feature leading uppercase letters of verbs and key nouns (“Gut Virome Analysis..”). In most other references, this is not so (e.g., 815). Please resolve this in a consistent way throughout the manuscript.

Also, the references in the Supplementary files are specified in a different way from what is done here. (Should journal names be abbreviated or not, for instance?) Again, consistency, please.

891, 907 and elsewhere. Species and genus names should be written in italics. No exceptions allowed.

All figures and all supplementary figures. Please revise the use of leading uppercase letters for headings. Figure 1 sports both “Small Intestine Length” and “Fecal fungal growth (week 7)”. This gives a sloppy impression. Please resolve this in a consistent way throughout the manuscript.

Figure 1 legend and elsewhere. “12 bacteria” (and so on). But what species were these? Please provide this information somewhere.

Figure 3 and elsewhere. Please consider giving full species names here. Not all readers will know what “R_mucil...” is, particularly since no lookup table is provided.

Figure 3 also begs this question: since the authors explored a less-well explored habitat for fungal diversity, were any of the “top 50 most wanted fungi” (<https://mycokeys.pensoft.net/articles.php?id=7553> ; <https://unite.ut.ee/repository.php>) found?

Reviewer #2 (Remarks to the Author):

In this manuscript van Tilburg Bernardes et al. have investigated the microbiome, metabolome, and immune effects of colonizing germ-free mice with a select group of bacteria and/or a select group of fungi. The approach has allowed the investigators to provide proof-of-concept data indicating that bacterial and fungal populations can have independent and interactive effects on intestinal and immune biology. While a generally descriptive study (we have no idea why observed effects happened or when or how they might be meaningful) the approach provides much food for thought. I have several specific comments:

1. The subtitle/statement “Intestinal physiology is adapted to bacterial, not fungal colonization” is overstated. What the investigators mean is that they observed that bacterial colonization induced some physiological changes in some parts of the gut that were not induced by yeast colonization, which is really interesting.
2. The abstract and subtitle statement “exclusive fungal colonization was protective against DSS-colitis” seems to be poorly substantiated by the data. As I am seeing it, the data show that the fungi the investigators selected did not, by themselves, exacerbate or protect from disease by most measures.
3. I am confused by the statistical comparisons. The use of letters to “indicate statistically significant differences” was tidy but left me wondering which comparisons were meant. For example, Fig. 1d – a & b are “statistically significant, but compared to what? I’m guessing the 10-fold reduction in yeast

colonies when bacteria are present is significant, but maybe the comparisons are to GF and B. It gets even more confusing when more comparisons are seemingly done (e.g. 1g).

4. In the sequencing analyses throughout, I don't understand why the investigators did not collapse ASVs identifying each organism. Normally, you don't know whether different ASVs represent different species or strains, so you have to keep them separate. But in this case the investigators know the identities of each species/strain introduced. There are not, for example, multiple strains of *Candida glabrata* present, so analyzing each *C. glabrata* ASV separately doesn't make sense. You know that when two different ASVs are attributed to *C. glabrata* that this is an artifact of sequencing and/or bioinformatics.

5. Did the investigators check by PCR whether sequencing was quantitative compared to PCR? Usually you can't do this, because you don't know all the organisms present, so you have to do sequencing to identify them and hope the sequencing is at least semiquantitative. In this case, the investigators know all of the organisms present.

6. I worry that the evaluations of "richness" measurements are misleading. There are simply 12 bacterial species/strains and/or 6 fungal species/strains present. If certain bacteria or fungi vanish in the sequencing of certain animals, we can be specific rather than referring to reductions in richness.

7. It is a little frustrating that the fungi used are not easily referenceable/acquired strains. An advantage of the gnotobiotic approach is that it can easily be fully replicated. But I wouldn't want to kill the study because of this.

8. Substantial care should be taken throughout the text to not make conclusions as to what bacteria or fungi are sufficient (or incapable) of doing. Only a subset of bacteria and fungi are employed here. This is great for proof-of-concept demonstrations, and I like the study for this. But we have to be careful about extrapolating too far. For example, the investigators conclude that "although fungi colonized the mouse gut, their presence was insufficient to induce any physiological or morphological changes in the ex-GF mice." What the investigators mean is that a mix of 6 common fungi were not able to induce the measured changes, while a mix of 12 bacteria were. Other fungi might, and other bacteria might not, be able to do this. This imprecision runs throughout the manuscript.

We sincerely thank the reviewers for their thorough review and constructive suggestions that have greatly improved our manuscript. Please note that all the mention of line numbers in this document are in reference to the manuscript word document **with changes**. New text is noted in blue, whereas deleted text is denoted with a strikethrough (~~abe~~).

COMMENTS FROM REVIEWER #1

1. This is an important and well-written experimental study on the role of fungi in mouse intestines. It is clear that a lot of work has gone into this study.

The authors would like to thank the reviewer for such a thorough and insightful review of the manuscript. We have done our best to fully respond to each of the points raised. Below please find our point-by-point response to each of the comments (which are shown in italics).

2. (Line 1) I'd prefer a more informative title. Right now, the title says nothing about the results.

We thank the reviewer for this point. We agree the title was quite general and did not highlight the main findings of the paper. We have changed the title to “Intestinal fungi are causally implicated in microbiome assembly and immune development in mice”.

3. (63-64) The authors are critical that other researchers routinely ignore fungi in microbiome studies, and I agree. Yet the authors fall into the same trap that everyone falls into – highlighting human-associated fungi as solely pathogenic. The authors thus give an unbalanced account of the role of fungi in human health and disease. See, e.g., <https://journals.plos.org/plospathogens/article?id=10.1371/journal.ppat.1002808> , <https://www.ncbi.nlm.nih.gov/pubmed/28102762> , and <https://www.ncbi.nlm.nih.gov/pmc/articles/PMC5840654/> for various non-pathogenic, probiotic, and evolutionary examples.

We also thank the review for this comment. We have revised the Introduction (lines 61-62, 66-71) to address this point, by adding the suggested articles to the manuscript. We have incorporated information on the different source of colonizing fungi that constitute the mycobiome and the use of fungi as probiotics to prevent intestinal infection and complications.

4. (78) Remarkably for a study on fungi, none of the authors seem to be a mycologist (even though Canada is full of mycologists). This shows over and over again in the manuscript. On this line, “Pichia kudriavzevii” is an obsolete synonym of “Issatchenkia orientalis”. See Index Fungorum (<http://www.indexfungorum.org/Names/Names.asp>).

We appreciate the reviewer’s comment regarding the importance of using the correct taxonomical names for fungal species. Indeed none of the authors are mycologists; however, there are strong microbiome scientists among the authors that are focused on expanding our understanding of all the components of the microbiome, not only bacteria. This work originated from a growing interest in the microbiology field to understand the contribution of fungal

colonizers in host immune modulation and potential causal implication in inflammatory pathologies. This study was designed based on literature that associated intestinal fungi to immune alterations, including observation from work published by the senior author (Marie-Claire Arrieta), in which there were strong associations between early life fungal alterations and infant susceptibility to allergic wheeze development by school age (PMID: 29241587).

In regards of use of an obsolete name for the yeast *Pichia kudriavzevii*, we thank you for bringing this to our attention, we have changed the name to *Issatchenkia orientalis* throughout the manuscript and figures.

We would like to thank the reviewer for bringing up this issue as it allowed us to find another misnomer that escaped our initial understanding of the fungal species in question. We searched for the source of the misnomer and came across the recent paper by Dr. Kenneth H. Wolfe et al (2018, PLoS Pathogens, PMID: 30024981). This study sequenced different environmental isolates of *I. orientalis* (*P. kudriavzevii*) and clinical isolates of *Candida krusei* to show that they are indeed the same species with genomes being 99.6% identical. *C. krusei* is the asexual (anamorph) form of *I. orientalis* (Kurtzman CP, Fell JW, Boekhout T, editors. The yeasts: a taxonomic study. 5th ed. London: Elsevier; 2011) and, therefore, we removed *C. krusei* from Supplementary Table 1, where we list all the yeast species used in this study.

5. (102) How can the authors know that all of the fungi in their study are “commensal”? This strikes me as very unlikely. It is also hard for me to assess, since the authors withhold most of their primary data (see below). If the authors want to classify the fungi recovered into functional guilds, then FUNGuild (<https://www.sciencedirect.com/science/article/abs/pii/S1754504815000847>) is readily available. I find it hard to come up with a reason why the authors chose not to use FUNGuild to build a functional profile of the fungi recovered.

We thank the reviewer for this comment. First, we want to clarify that the taxonomic and source information of all microbes used in our study were in Supplementary Table 1. It was never our intention to hide these data. Second, we agree with the reviewer that the word commensal was broadly and loosely used based on the fact that the yeast species chosen are normal colonizers of the mammalian intestinal tract (PMIDS: 29241587, 27618652, 16391120).

We do know that in our model system, none of the yeasts used to colonize these mice induced overt inflammation either in the lung or the gut, nor did they cause fungemia. This is based on histological findings in the gut and lung under normal conditions (naïve groups for gut and lung), as well as fungal cultures from kidney, which is a standard method to detect fungemia. Thus, fungi did not appear to infect and cause disease in our gnotobiotic system. With that said, we think the reviewer is correct that proving a commensal relationship between symbionts is very hard to do, and we simply do not know for sure if this is the case. Therefore, to address the reviewer's comment, we have removed the word “commensal” from the manuscript and replaced it by “colonizer”.

In regard to the use of FUNGuild, we fully agree that FUNGuild is a remarkable tool for the taxonomic clustering of complex microbial communities by functional groups. After using this tool to obtain functional profiles of the 5 yeasts species used in our study, we found that, due to their phylogenetic closeness, this tool yielded very similar profiles to all species chosen. This has now been added to Supplemental Table 1.

6. (180) *The ITS region does not formally qualify as a “gene”, so on this line, “gene sequencing” is not formally correct. I propose “marker sequencing”.*

We appreciate the comment from the reviewer. We have addressed this issue and have replaced the word “gene” throughout the text to the appropriate “genetic region” or “marker” as proposed.

7. (188) *The abbreviation “PERMANOVA” is defined twice in the manuscript. I’d say once is enough?*

We thank the reviewer for noticing this oversight on our part. We have removed the second definition to PERMANOVA (line 789)

8. (226) *The authors withhold the list of fungal [and, for that matter, bacterial] species recovered (presumably for a second publication). Not releasing the full set of results is not acceptable if you ask me. Please provide:*

- * a summary pie chart showing the taxonomic affiliations of all fungi [and bacteria] recovered*
- * a FUNGuild chart of the functional guilds of the fungi recovered*
- * a supplementary item with all ASVs and their taxonomic affiliation, including the DOI of each ASV as assessed through comparison to the UNITE database (<https://unite.ut.ee/repository.php>). Most of the ASVs of the authors will presumably lack full species names, so providing a DOI is the only way to make the taxonomic results comparable across studies and time. Failure to do so would compromise the scientific reproducibility of the results.*

We wondered if perhaps the reviewer was not given access to the supplementary material because our Supplemental Table 1 had this important information all along. With that said, we appreciate the reviewers comment to further increase the information we initially provided. Supplemental Table 1 now has information on functional profiles for all yeast species from FUNGuild. We have also now provided additional Source Data file containing excel spreadsheets with complete and merged ASV-level information. Complete ASV tables include all ASVs generated by DADA2, whereas merged ASV tables include the merged ASVs with the same species name that explained more than 99% of the abundance. Because of our gnotobiotic approach using well-known yeasts species with annotated genomes in the UNITE database, we were able to obtain full species names for the vast majority of the ASVs. Our complete ASV table now also includes the complete sequence, which can be used by readers to compare across

studies and time. We thank the reviewer for encouraging this.

9. (230) “*taxa*” should be “*taxon*”. Singular noun. Or are the authors somehow implying that this name is non-monophyletic? If so, please back it with at least one solid recent reference.

Thank you. We have corrected to the singular noun (line 245).

10. (237) What is the reader supposed to get from “(bicor method, FDR corrected)” at this stage in the manuscript?

Thank you. We have now fully defined the BiCOR abbreviation. The Biconjugate A-Orthogonal Residual (BiCOR) method is an algorithm for solving nonsymmetric linear systems (Zhao L. *et al.* Comput Math with Appl 2013 <https://www.sciencedirect.com/science/article/pii/S0898122113004884>). The appropriate definition was included in the text in line 257.

11. (238-242) Please write the species names out in full. Most readers will have no clue what species are intended otherwise.

Thank you for this comment. The full names for all five yeast species included in this study were displayed at first appearance in the manuscript, as well in Supplementary Table S1. We have now also included full species names in the methods section. Lines 536-540.

12. (258) “Heat-map” > “The heap-map” ?

Thank you. We have corrected it in the text. Line 279.

13. (271) Most of the time, the authors use the Oxford comma when it comes to enumerations (e.g., 327 and 329). But on this line and, e.g., 272, they don’t. Please resolve this in a consistent way throughout the manuscript.

We appreciate the reviewer for noticing that we value the use of the Oxford comma, yet missed it in some places. We carefully went over the whole manuscript to correct this.

14. (279) “yielded specific grouping of the mice” sounds awkward. Please revise for clarity.

We agree that this sentence could have been improved and have revised it in lines 297-299, to say “we did not observe any specific grouping of the mice based on the metabolome of the small intestinal luminal content”.

15. (369-373) *This part is very interesting. Should it be given a more prominent role in the manuscript (including the abstract)?*

We thank the reviewer for his/her interest in these results. We agree that bacteria (but not fungi) reducing airway inflammation in this gnotobiotic model is a remarkable observation and we have added the distinction between bacteria and fungi in the revised abstract.

We further highlighted the fact that animals in both the B group and BY group displayed reduced OVA-induced airway inflammation, even if BY animals harbour fungi that have been linked to allergic airway inflammation before (PMIDs: 15321991, 15618138, 24439901). We hypothesize that because fungi are substantially outnumbered by bacteria, ecosystem perturbances, either through the use of antibiotics or antifungals, are required for fungi to exacerbate airway responsiveness to allergic challenges (lines 388-393).

16. (385) *Can this really be true? After all, the fact that there are fungi that are specialized in colonizing mammalian intestines has been known for 100+ years?*

(<https://en.wikipedia.org/wiki/Neocallimastigomycota> ; <https://mycokeys.pensoft.net/article/28337/>)

We also do not believe this to be true, but it was suggested in this paper from Dr. Petrosino's group (PMID: 29600282), in which they found similar fungi in the diets, saliva, and feces of humans. Their interpretation of this is that we are constantly exposed to fungi, which are transiently passing through the gastrointestinal (GI) tract but not real dwellers. We disagree with this view, and our data strongly indicates that fungi can actively colonize the murine intestinal tract. We were able to not only sequence but also culture these fungi weeks after initial colonization, and confirm they are metabolically active members of the intestinal microbiome.

17. (395) *Again the authors focus solely on the negative aspects of fungi. There are many examples of symbioses between fungi and bacteria – such as lichens and, e.g., <https://www.ncbi.nlm.nih.gov/pmc/articles/PMC3232736/> - but the authors conveniently ignore those. Also, please provide a (modern) reference to back the claim that most fungal-bacterial interactions in the soil take the form of “competition and antagonism”. The authors will probably struggle to come up with such a reference.*

We appreciate the reviewer's comment and have now expanded from mostly antagonistic bacterial-fungal interactions. It was never our intention to suggest most fungal-bacterial interactions are competitive or antagonistic. We have edited the text to better indicate that in our setting we observe apparent antagonistic and synergistic correlations between bacteria and fungi. We also thank the reviewer for sharing the review paper by Dr. Markus Künzler (PMID: 22126995). We have further inserted a sentence citing both beneficial and antagonistic bacterial-fungal interactions, and further associated our specific observations to other previous observations described in other ecosystems. (lines 424-428).

18. (435) Please clarify the “determining...” statement. I have a hard time making sense out of it.

We have changed this sentence to “Our results support previous work showing that bacterial colonization is essential for the development of DSS-colitis⁴⁰, and expand current knowledge by demonstrating that exclusive colonization with this fungal consortium delays and ameliorates overt colitis.” in lines 469-473.

19. (462) The “sp.” should not be given in italics.

This has now been corrected through the text, thank you (lines 81 and 499).

20. (526) “For 16S” > “For the 16S” ?

This has also been corrected. Line 567.

21. (531) “For fungal” > “For the fungal” ?

This has also been corrected. Line 572.

22. (629) Of all 16S regions the authors could have chosen, the V4 region is the least informative one from a taxonomic point of view (see Fig. 1 in <https://www.sciencedirect.com/science/article/pii/S0167701210002745?via=ihub>). Why was this poorly performing region chosen to begin with?

In our gnotobiotic setting, in which animals were colonized with 12 known bacterial species with genomes that have been fully annotated, we did not experience any limitations sequencing the V4 region. Combined with the use of the latest release of the SILVA database, we were able to obtain species resolution for the bacterial species used. We agree that other V regions perform better, especially in other microbial environments, but for our experimental setting this region performed very well.

23. (629) There are no V regions in the fungal ITS region. So what the authors mean by “V3-V4” here is beyond me. Is this a simple copy-and-paste mistake from the 16S part?

Thank you for pointing out this oversight on our part, this has been corrected (line 670). We used the primer pair ITS86F/ITS4 to amplify the ITS2 region of the ITS marker, as previously described (PMID: 24933453).

24. (630) Like I said before, the ITS region does not formally qualify as a “gene”. Please use “marker” or “genetic marker” instead.

The proper term for this genetic marker has been corrected throughout the text (lines 193 and 671).

25. (647) *The metabarcoding part leaves a lot to be desired in terms of reproducible science. Please provide considerably more information here. What pipeline was used to process the raw data? What ASV parameters were used? Why wasn't LULU (<https://www.nature.com/articles/s41467-017-01312-x>) used? What taxonomic reference databases were used? Which software tools were used, and what were their version numbers? Where can the reader access the raw sequence data and the OTU tables? I presume this is the first time the authors report on metabarcoding results – please see, e.g., <https://academic.oup.com/gigascience/article/4/1/s13742-015-0074-5/2707574> and <https://nph.onlinelibrary.wiley.com/doi/10.1111/j.1469-8137.2011.03755.x> on how to do it.*

Thank you for your suggestion on this matter. Our method section initially included that we used DADA2 as our pipeline to process raw sequence data for both 16S and ITS2 datasets. We had now also provided the reference for these pipelines, which include step-by-step tutorials on the bioinformatics steps to produce ASVs for bacteria or fungal sequences, including the databases for taxonomy assignments. We have improved our description of these methods by now including the references to the databases themselves, the url for the pipeline tutorials, and the versions of the R packages used. Lines 686-695.

In regards to LULU, this tool is designed for studies in which the community identity is large and not known (e.g. soil or gut microbial communities), and it is based on clustering methods. For our study, in which we knew exactly which species were introduced to these mice, we opted for a non-clustering approach. Biologically, DADA2 is a tool that aims to identify the exact sequence variants that exist in communities, where taxa can belong to the same strain/species but developing mutations in their genetic sequence of marker genes, therefore also providing information on the potential genetic changes of the community members. This was also important for us because we will be looking at some of these changes in future studies. Broad advantages of using DADA2 instead of cluster-based methods that generate OTUs (including the clustering algorithm used in LULU) can be found in this ISME J perspective article by Callahan, McMurdie and Holmes (PMID: 28731476).

26. (734) *“R package” > “The R package”. Also, please provide the version number of R and all other software tools and packages used in the study. Failure to do so would compromise scientific reproducibility (which is already quite low in this study, owing to the authors’ withholding the raw data, the OTU (ASV) tables, the full taxonomic results, and so on).*

Thank you for noticing this. This has all been corrected (line 802). The additional requested information has been added in lines 686-695

27. (748) *“fungi” > “fungal”*

Thank you for noticing this. This has all been corrected. Line 819.

28. (778) *What, exactly, was deposited here? Please deposit all relevant files, including the raw sequence data, in ENA or SRA. And please follow standards by also providing the deposition accession number.*

All raw 16S and ITS2 sequences and supporting metadata have been deposited to ENA (<https://www.ebi.ac.uk/ena/browser/view/PRJEB35163>). The study accession numbers (PRJEB35163/ERP118175) are now included in line 850-851. ASV tables and other relevant metadata are also included in the Source Data file provided.

29. (818 and elsewhere) *The list of references is heterogeneously specified. In this particular reference, article titles feature leading uppercase letters of verbs and key nouns (“Gut Virome Analysis..”). In most other references, this is not so (e.g., 815). Please resolve this in a consistent way throughout the manuscript.*

Thank you for noticing this. This has all been corrected.

29. *Also, the references in the Supplementary files are specified in a different way from what is done here. (Should journal names be abbreviated or not, for instance?) Again, consistency, please.*

We thank the reviewer for this observation and we have corrected the references in the supplementary file to be consistent with the main text file.

30. (891, 907 and elsewhere) *Species and genus names should be written in italics. No exceptions allowed.*

We thank the reviewer for noticing that. We did not notice that Endnote automatically does this. This has now been corrected.

31. *All figures and all supplementary figures. Please revise the use of leading uppercase letters for headings. Figure 1 sports both “Small Intestine Length” and “Fecal fungal growth (week 7)”. This gives a sloppy impression. Please resolve this in a consistent way throughout the manuscript.*

We thank the reviewer for such a meticulous review of the manuscript and for identifying all these details that have escaped our several rounds of revisions. This has now been revised throughout the figures.

32. *Figure 1 legend and elsewhere. “12 bacteria” (and so on). But what species were these? Please provide this information somewhere.*

This was always included in Supplementary Table S1, and is now also included at the beginning of the methods section (lines 536-540).

33. *Figure 3 and elsewhere. Please consider giving full species names here. Not all readers will know what “R_mucil...” is, particularly since no lookup table is provided.*

We appreciate the reviewer’s comment. Besides including full species names in Supplementary Table S1 and at the beginning of the methods section (lines 536-540), we are using abbreviated species names in the figures (ie. *C. albicans* instead of *Candida albicans*). This abbreviation is necessary for space reasons, though we are confident that having the full species names in two other locations in the paper will be sufficient.

34. *Figure 3 also begs this question: since the authors explored a less-well explored habitat for fungal diversity, were any of the “top 50 most wanted fungi” (<https://mycokeys.pensoft.net/articles.php?id=7553> ; <https://unite.ut.ee/repository.php>) found?*

We thank the reviewer for this comment and for sharing the manuscript by R. Henrik Nilsson *et al.* It was an interesting read addressing commonly found unidentified sequences in complex environmental communities, and it will improve our other ongoing study in complex fungal communities. However, for this study we used a gnotobiotic approach of five known fungal species being included in our communities, and we did not obtain any sequences that were unable to be assigned at least to a genus level. Therefore, none of the “top 50 most wanted fungi” could be detected.

COMMENTS FROM REVIEWER #2:

In this manuscript van Tilburg Bernardes et al. have investigated the microbiome, metabolome, and immune effects of colonizing germ-free mice with a select group of bacteria and/or a select group of fungi. The approach has allowed the investigators to provide proof-of-concept data indicating that bacterial and fungal populations can have independent and interactive effects on intestinal and immune biology. While a generally descriptive study (we have no idea why observed effects happened or when or how they might be meaningful) the approach provides much food for thought. I have several specific comments:

The authors would like to thank the reviewer for such an insightful review of the manuscript. We have done our best to fully respond to each of the points raised. Below please find our point-by-point response to each of the comments (which are shown in italics).

1. The subtitle/statement “Intestinal physiology is adapted to bacterial, not fungal colonization” is overstated. What the investigators mean is that they observed that bacterial colonization induced some physiological changes in some parts of the gut that were not induced by yeast colonization, which is really interesting.

We would like to thank the reviewer for their comment. We have edited several sentences to highlight the proof-of-concept nature of our work and to remove this overstatement (lines 116, 143, 166-168, 179, 303-304, 312, 396, 431, 437-439, 471). With that said, it is important to note that we used very common members of the fungal communities found in the human GI tract, including *C. albicans*. While our work does not rule out that other common fungi may elicit physiological changes, our data does suggest that these very common members did not elicit any noticeable changes in several regions of the mouse GI tract (ileum, caecum, and colon). We have made this clear in the revised paper.

2. The abstract and subtitle statement “exclusive fungal colonization was protective against DSS-colitis” seems to be poorly substantiated by the data. As I am seeing it, the data show that the fungi the investigators selected did not, by themselves, exacerbate or protect from disease by most measures.

We also thank the reviewer for this comment. We have changed this sentence to “Our results support previous work showing that bacterial colonization is essential for the development of DSS-colitis⁴⁰, and expand current knowledge by demonstrating that exclusive colonization with this fungal consortium delays and ameliorates overt colitis.” (lines 469-473). We have also addressed the reviewers comment and changed the wording in the abstract. We believe this better aligns with our data and thank the reviewing for noting this.

3. I am confused by the statistical comparisons. The use of letters to “indicate statistically significant differences” was tidy but left me wondering which comparisons were meant. For example, Fig. 1d – a & b are “statistically significant, but compared to what? I’m guessing the 10-fold reduction in yeast colonies when bacteria are present is significant, but maybe the comparisons are to GF and B. It gets even more confusing when more comparisons are seemingly done (e.g. 1g).

We appreciate this comment. The idea behind using letters was to make the graphs tidier and more legible, as multiple statistically significant differences were commonly among the bars. The alternative of this led to using multiple lines and stars (up to 14 for one plot at times), which rendered the statistical analyses illegible. For all the graphs, comparisons are done between all groups. The alphanumeric system to denote posthoc statistical analysis helps to reduce complexity, but it is certainly less intuitive. Whenever bars have different letters, they are significantly different from each other. The same rule applies if two letters are necessary above a single bar.

For example, for Fig 1g (see below), the B group, which has an ‘bc’ above the bar, is not significantly different to groups with marked with ‘b’ or ‘c’ above the bars. Also, if bars do not

have a letter on top of them (i.e. Fig 1d, for GF and B) it is because all values were zero, and the standard deviation and inferential statistics cannot be calculated. To better explain our representation of statistically significant differences we included a sentence at the end of ‘Data Visualization and Statistical Analysis’ methods section (line 811-814) as well as at the end of the legend to Figure 1. We hope this added statement better explains the approach used to all readers.

4. In the sequencing analyses throughout, I don't understand why the investigators did not collapse ASVs identifying each organism. Normally, you don't know whether different ASVs represent different species or strains, so you have to keep them separate. But in this case the investigators know the identities of each species/strain introduced. There are not, for example, multiple strains of *Candida glabrata* present, so analyzing each *C. glabrata* ASV separately doesn't make sense. You know that when two different ASVs are attributed to *C. glabrata* that this is an artifact of sequencing and/or bioinformatics.

We truly appreciate the reviewer's suggestion and have incorporated it to our community-level analyses. We agree this makes a lot more sense considering our gnotobiotic experimental setting. Our new approach first selected the ASVs from the 16S and ITS2 datasets that explained >99% of community abundance. This resulted in the merging of 22 bacterial ASVs and 16 fungal ASVs. We have also provided excel spreadsheets including complete ASV tables as well as merged abundance tables for the 16S and ITS2 datasets in the Source Data file.

Interestingly, correlation analyses between fungi and bacteria failed to find any significant correlations once the taxa had been merged. Because so many correlations were found at the ASV-level, which matched the overall changes in abundance observed, we chose to leave this analysis at the ASV-level. We hypothesize that DADA2, which is a tool that aims to identify the exact sequence variants that exist in communities, and is sensitive to mutations in the genetic sequence of marker genes, may reflect biologically relevant changes to community members. Alternatively, this may be solely a result of bioinformatic and sequencing artifacts, although one would expect these to occur randomly. We are currently studying this in the lab for a future study and publication.

5. Did the investigators check by PCR whether sequencing was quantitative compared to PCR? Usually you can't do this, because you don't know all the organisms present, so you have to do sequencing to identify them and hope the sequencing is at least semiquantitative. In this case, the investigators know all of the organisms present.

This was another very valuable suggestion, thank you. We were also intrigued by this question and decided to check if changes in abundance could be quantified by qPCR. For this, we chose to carry out specific qPCR of *C. albicans* and *R. mucilaginosa*. As now seen in Fig. 3e, we confirmed a significant reduction in *C. albicans* gDNA only in the antifungal-treated group, while *R. mucilaginosa* gDNA signal was mostly detected only in groups where *C. albicans* was depleted (Supplementary Fig. 3b-c). These results support the findings of our sequencing approach and provide added confidence of the ecological shifts we reported. Description of the new methodology used, including primer sequences, are in lines 697-714, and Supplementary Table 1.

6. I worry that the evaluations of “richness” measurements are misleading. There are simply 12 bacterial species/strains and/or 6 fungal species/strains present. If certain bacteria or fungi vanish in the sequencing of certain animals, we can be specific rather than referring to reductions in richness.

We thank the reviewer for this insightful comment, and indeed once we merged the community ASVs we noticed that DADA2 likely overestimates number of species because there was a reduction in the Chao1 index for both bacteria and fungi, but especially for fungi (see figures

below). Nevertheless, we were still able observed similar trends as before. These include a reduced bacterial Chao1 index in group co-colonized with fungi (BY) or treated with antibiotics (Abx) at week 4, as can be seen in the figure below. At week 9, we observed an increase in Chao1 index for BY group, while BY+Abx was unable to recover from antimicrobial treatment. Similarly, we observe a reduced fungal Chao1 index in animals exclusively colonized with fungi, both before and after merging ASVs originated by DADA2.

Chao1 before merging ASVs

Chao1 after merging ASVs

Shannon before merging ASVs

Shannon after merging ASVs

■ B ■ Y ■ BY ■ BY+Abx ■ BY+Afx

While the richness results were consistent to what we have reported in the previous version of this paper, following a group discussion we decided to focus our discussion on alpha-diversity to the results from the Shannon index. We believe this index is a more robust measure of community changes, especially in the context of a community of reduced diversity. The results for the Shannon index did not change before or after merging the ASVs at the species level. Thus, we have chosen to include Shannon index plots in the new version of Figure 3.

7. It is a little frustrating that the fungi used are not easily referenceable/acquired strains. An advantage of the gnotobiotic approach is that it can easily be fully replicated. But I wouldn't want to kill the study because of this.

We appreciate the reviewers comment and, to some extent, we agree with this view. Ideally, we could have used fully referenced strains for all fungi (we did use DSMZ strains for *Issatchenkia orientalis* and *Rhodotorula mucilaginosa* and provided DSMZ strain number for these in Supplementary Table 1). However, our rationale to choose these species was to use taxa that were previously detected in infant cohort studies as linked to atopy or asthma risk. As such, we decided it would be best to use strains isolated from humans, and obtained the *Candida* species from a clinical collaborator. To improve the ability of other researchers to replicate our work, we have added a sentence in the Data and materials availability section noting that we are open to share these strains with those that request them (lines 851-852).

8. Substantial care should be taken throughout the text to not make conclusions as to what bacteria or fungi are sufficient (or incapable) of doing. Only a subset of bacteria and fungi are employed here. This is great for proof-of-concept demonstrations, and I like the study for this. But we have to be careful about extrapolating too far. For example, the investigators conclude that "although fungi colonized the mouse gut, their presence was insufficient to induce any physiological or morphological changes in the ex-GF mice." What the investigators mean is that a mix of 6 common fungi were not able to induce the measured changes, while a mix of 12 bacteria were. Other fungi might, and other bacteria might not, be able to do this. This imprecision runs throughout the manuscript.

We appreciate the reviewer voicing this concern and we agree that the conclusions obtained were related to the specific microbial consortium we used. We also agree that some of our initial statements are overstated. We have carefully reworded our conclusions and results throughout the manuscript (lines 116, 143, 166-168, 179, 303-304, 312, 396, 431, 437-439, 471).

REVIEWERS' COMMENTS:

Reviewer #1 (Remarks to the Author):

I am by and large happy with the revised manuscript. It certainly reads very well, too. Some minor/cosmetic issues remain.

32, 33. Is it wise to include un-defined abbreviations in the Abstract?

35-37. This is a wise conclusion. I hope other researchers will heed.

58. "focused of bacteria" > "focused on bacteria"

98. "neither can" > "nor can" ?

100. You would not say "a plants-related question" (but rather "a plant-related question"). So would you say "fungi-induced" or "fungus-induced"? I would lean towards the latter.

102. "specific pathogen free, SPF" > "specific pathogen free (SPF)". That is, at least, how you defined abbreviations elsewhere in the manuscript.

112. Will all readers know what is meant by "ex-germ-free mice"? I am less sure, so why not explain this concept briefly.

120-121. Since bacteria are "B", shouldn't fungi be "Y" rather than "-Y"?

130. "genes specific" > "gene-specific" ?

132. "Copy number" is used in an unfortunate way here. It normally refers to the number of (here) 18S gene copies per cell (genome). A fungal genome would thus have more 18S (SSU) copies than a bacterial genome would 16S. Anyway, this was not what the authors examined, so maybe they should consider clarifying the text a bit.

165. "exacerbated s disease" > "exacerbated disease"

188. "the internal transcribed spacer" > "the nuclear ribosomal internal transcribed spacer" in the interest of clarity.

235. "reflect" > "reflected" ?

242. The abbreviation "ASV" is defined only much later in the manuscript.

271. Is "indicating" too strong a word here? Is "hinting at" or "possibly hinting at" better?

277. "in citric" > "in the citric" ?

332 and 352. "associated to" > "associated with" ? or "ascribable to" ?

334. "species" can be removed. Or the sentence can be rewritten to start "The species ..."

360. I would go "several bacterial species" or "a number of bacterial species"

387. "resulting results" > "resulting"

435. "making them likely a" – sounds awkward to me. And what does "them" refer to anyway?

472. "in in" > "in".

511. “prevent its” > “prevent their” ?

523, 525. The “and” should not be given in italics.

544. Is a reference needed for this method? Either that or some more detail, I’d say.

661. “In house” > “In-house” ?

663. “-normalized” > “normalized” ?

703. In my world, “IMC” means “International Mycological Congress”. But I take it that the authors intend a different interpretation here.

751. “Experimental” > “An experimental” ?

774. “taxonomical” > “taxonomic”

782. “immune features” > “and immune features” ?

Figure 1g-h – different font size in header, right? Looks funny.

Figure 3 legend. “by BiCOR” > “by the BiCOR” or “by a BiCOR” ?

Supplementary Table 4. Why the mix of uppercase and lowercase names? What is the difference between “N-formyl-L-methionine” and “5-HYDROXY-L-TRYPTOPHAN” (and so on)? If a difference is intended, then it needs to be explained to the reader.

Reviewer #2 (Remarks to the Author):

In general, authors' responses to the prior review are sufficient. It's an interesting set of observations that should attract the attention of many in the field. I do, however, have a couple remaining concerns regarding interpretation and discussion.

I don't understand the investigators' insistence that "... bacterial colonization is essential for the development of DSS-colitis, and expand current knowledge by demonstrating that exclusive colonization with this fungal consortium delays and ameliorates overt colitis." Exclusive colonization with fungi did almost nothing to the DSS colitis (neither "prevented overt" disease or exacerbated it). The only statistically significant change noted in fungi-only animals is some reduced shortening of the colon compared to GF animals. ALL OTHER measures offered are not affected. Thus, a conclusion more along the lines of "had little or no effect" would seem to make more sense. The newer text in the abstract is clearer on this conclusion.

From the response: "Interestingly, correlation analyses between fungi and bacteria failed to find any significant correlations once the taxa had been merged." Doesn't this suggest then that correlations are unlikely? ASV level associations don't make sense in this experimental setting and must be viewed with a lot of skepticism. It is extraordinarily unlikely that meaningful ITS sequence variant strains emerged during the short time course of these experiments. If the authors insist on retaining data in 3f, they must at least acknowledge/discuss in the text that these associations were not noted when all ASVs attributed to each known organism were combined.

Did the “microbial-immune” correlation analysis also fail if not done with ASVs (fig 5d)? If so, the same concerns apply, and if not, the data should be shown using the total data for each known species.

We sincerely thank the reviewers for another careful review of the manuscript. We believe the latest changes have improved the quality of the work presented. Please note that all line numbers included in this document are in reference to the manuscript word document **with changes**. New text is noted in blue, whereas deleted text is denoted with a strikethrough (~~abe~~).

COMMENTS FROM REVIEWER #1

1. I am by and large happy with the revised manuscript. It certainly reads very well, too. Some minor/cosmetic issues remain.

The authors would like to thank the reviewer for such a positive feedback and another through review of the manuscript. We have addressed most of the points raised. Below please find our point-by-point response to each of the comments (which are shown in italics).

2. (lines 32-33) Is it wise to include un-defined abbreviations in the Abstract?

We have addressed this issue and removed the undefined abbreviations from the abstract.

3. (35-37) This is a wise conclusion. I hope other researchers will heed.

Thank you.

4. (58) “focused of bacteria” > “focused on bacteria”

We have corrected it in the text (line 58).

5. (98) “neither can” > “nor can” ?

This has also been corrected in the text (line 98).

6. (100) You would not say “a plants-related question” (but rather “a plant-related question”). So would you say “fungi-induced” or “fungus-induced”? I would lean towards the latter.

Thank you for noticing this mistake. We have replaced for “fungus-induced” as reviewer suggested (line 100).

7. (102) “specific pathogen free, SPF” > “specific pathogen free (SPF)”. That is, at least, how you defined abbreviations elsewhere in the manuscript.

The words “*specific pathogen free, SPF*” are inside of parenthesis (line 101-102), thus instead of adding another set of brackets, we have opted for having it following a semicolon (we eliminated the comma) to keep in line with the formatting across the entire manuscript.

8. (112) *Will all readers know what is meant by “ex-germ-free mice”? I am less sure, so why not explain this concept briefly.*

Thanks for noting this, to prevent any confusion from this term we have eliminated the “ex” and left the phrase as “germ-free mice colonized with defined consortia of either bacteria, fungi, or both.” (lines 113-114). Full description of our colonization approach is included in the methods.

9. (120-121) *Since bacteria are “B”, shouldn’t fungi be “Y” rather than “-Y”?*

Thank you. This has been corrected (line 122).

10. (130) *“genes specific” > “gene-specific” ?*

This has also been corrected (line 131).

11. (132) *“Copy number” is used in an unfortunate way here. It normally refers to the number of (here) 18S gene copies per cell (genome). A fungal genome would thus have more 18S (SSU) copies than a bacterial genome would 16S. Anyway, this was not what the authors examined, so maybe they should consider clarifying the text a bit.*

We have changed “copy number” to “target DNA” to clarify what was measured via qPCR (lines 133-134). We have also deleted another instance where “copy number” was used (line 135).

12. (165) *“exacerbated s disease” > “exacerbated disease”*

Thank you, this has been corrected (line 168).

13. (188) *“the internal transcribed spacer” > “the nuclear ribosomal internal transcribed spacer” in the interest of clarity.*

This has been corrected (line 191).

14. (235) *“reflect” > “reflected” ?*

This has also been corrected (line 239).

15. (242) *The abbreviation “ASV” is defined only much later in the manuscript.*

We thank the reviewer for noticing this. We have moved the full definition from lines 743-705 to here (line 248).

16. (271) *Is “indicating” too strong a word here? Is “hinting at” or “possibly hinting at” better?*

We agree. We have changed to “hinting” (line 290).

17. (277) *“in citric” > “in the citric” ?*

This has been corrected (line 296).

18. (332-352) *“associated to” > “associated with” ? or “ascribable to” ?*

“Associated to” has been changed to “ascribable to” (line 353) and “associated with” (line 363).

19. (334) *“species” can be removed. Or the sentence can be rewritten to start “The species ...”*

The word “species” was removed (line 354).

20. (360) *I would go “several bacterial species” or “a number of bacterial species”*

We changed the sentence to “several bacterial species” (line 382).

21. (387) *“resulting results” > “resulting”*

Thank you. This has been now corrected (line 409).

22. (435) *“making them likely a” – sounds awkward to me. And what does “them” refer to anyway?*

We agree, and we have changed this to “rendering fungi as a decisive factor...” (line 458).

23. (472) “*in in*” > “*in*”.

Thank you, this has been corrected (line 495).

24. (511) “*prevent its*” > “*prevent their*” ?

This has been corrected (line 535).

25. (523-525) *The “and” should not be given in italics.*

Thank you for noticing this. This has all been corrected (lines 547 and 549).

26. (544) *Is a reference needed for this method? Either that or some more detail, I'd say.*

We have added more details to the Methods section (lines 567-573).

27. (661) “*In house*” > “*In-house*” ?

This has been corrected (line 686).

28. (663) “*-normalized*” > “*normalized*” ?

This has also been corrected (line 692).

29. (703) *In my world, “IMC” means “International Mycological Congress”. But I take it that the authors intend a different interpretation here.*

We thank the reviewer for noticing this. “IMC” stands for International Microbiome Centre, and its acronym was first introduced in line 542.

30. (751) “*Experimental*” > “*An experimental*” ?

We have changed the highlighted to “An experimental” (line 782).

31. (774) “*taxonomical*” > “*taxonomic*”

This has been corrected (line 806).

32. (782) “immune features” > “and immune features” ?

Thank you for noticing this. This has now been corrected (line 814).

33. (Figure 1g-h) different font size in header, right? Looks funny.

We thank the reviewer for noticing this. This has been reviewed throughout the figures and we believe all font sizes match now.

34. (Figure 3 legend) “by BiCOR” > “by the BiCOR” or “by a BiCOR” ?

Thank you for noticing that. We have changed the figure legend to include “by the BiCOR method”, as suggested by the reviewer. We have also changed that in Fig 5 for consistency.

35. (Supplementary Table 4) Why the mix of uppercase and lowercase names? What is the difference between “N-formyl-L-methionine” and “5-HYDROXY-L-TRYPTOPHAN” (and so on)? If a difference is intended, then it needs to be explained to the reader.

We thank the reviewer noticing this, which has all been corrected in Supplementary Tables 4 and 5.

COMMENTS FROM REVIEWER #2

1. In general, authors’ responses to the prior review are sufficient. It’s an interesting set of observations that should attract the attention of many in the field. I do, however, have a couple remaining concerns regarding interpretation and discussion.

We thank the reviewer for the careful review of this manuscript and for the very stimulating comments and suggestions, all of which have been taken into account in this submission.

2. I don’t understand the investigators’ insistence that “... bacterial colonization is essential for the development of DSS-colitis, and expand current knowledge by demonstrating that exclusive colonization with this fungal consortium delays and ameliorates overt colitis.” Exclusive colonization with fungi did almost nothing to the DSS colitis (neither “prevented overt” disease or exacerbated it). The only statistically significant change noted in fungi-only animals is some reduced shortening of the colon compared to GF animals. ALL OTHER measures offered are not affected. Thus, a conclusion more along the lines of “had little or no effect” would seem to make more sense. The newer text in the abstract is clearer on this conclusion.

We thank the reviewer for bringing this concern and we are glad that the abstract information was clearer on this conclusion. We have expanded this in other sections of the manuscript to state that exclusive fungal colonization “had little to no effect on eliciting overt colitis”, as proposed (line 480-481).

3. From the response: “Interestingly, correlation analyses between fungi and bacteria failed to find any significant correlations once the taxa had been merged.” Doesn’t this suggest then that correlations are unlikely? ASV level associations don’t make sense in this experimental setting and must be viewed with a lot of skepticism. It is extraordinarily unlikely that meaningful ITS sequence variant strains emerged during the short time course of these experiments. If the authors insist on retaining data in 3f, they must at least acknowledge/discuss in the text that these associations were not noted when all ASVs attributed to each known organism were combined. Did the “microbial-immune” correlation analysis also fail if not done with ASVs (fig 5d)? If so, the same concerns apply, and if not, the data should be shown using the total data for each known species.

We would like to once again appreciate the reviewer for making this highly insightful comment. In an attempt to run correlation analysis between merged ASVs and immune parameters, as was suggested, we detected a small mistake in the R script that had incorrectly yielded no significant correlations with the merged ASVs. Upon noticing this, we found not only strong correlations between microbial species (merged ASVs) and the immune parameters, but also between bacteria and fungi. Thus, we incorrectly reported in our previous submission that there were no significant correlations, when in fact there were, and these are very much in line with the ASV-level correlations reported previously. This has added confidence in our previous results and conclusions. We have therefore updated the correlation figures with merged ASVs at the species level (Fig 3 and 5) and changed the text file accordingly (Lines 246-273 and 349-375) to reflect the correct correlations.